# Evolutionary dynamics of any multiplayer game on regular graphs

Chaoqian Wang ®[1] ✉, Matjaž Perc ®[2,3,4,5] & Attila Szolnoki ®[6]

Multiplayer games on graphs are at the heart of theoretical descriptions of key evolutionary processes that govern vital social and natural systems. However, a comprehensive theoretical framework for solving multiplayer games with an arbitrary number of strategies on graphs is still missing. Here, we solve this by drawing an analogy with the Balls-and-Boxes problem, based on which we show that the local configuration of multiplayer games on graphs is equivalent to distributing $k$ identical co-players among $n$ distinct strategies. We use this to derive the replicator equation for any $n$-strategy multiplayer game under weak selection, which can be solved in polynomial time. As an example, we revisit the second-order free-riding problem, where costly punishment cannot truly resolve social dilemmas in a well-mixed population. Yet, in structured populations, we derive an accurate threshold for the punishment strength, beyond which punishment can either lead to the extinction of defection or transform the system into a rock-paper-scissors-like cycle. The analytical solution also qualitatively agrees with the phase diagrams that were previously obtained for non-marginal selection strengths. Our framework thus allows an exploration of any multi-strategy multiplayer game on regular graphs.

Multi-strategy evolutionary dynamics in nature often lead to diverse and complex phenomena, such as cyclic dominance that is captured by the well-known rock-paper-scissors game[1]. Experimental evidence from diverse contexts, ranging from the three-morph mating system of the side-blotched lizard[2] and *Escherichia coli* populations[3], to human economic behaviors[4], demonstrates the occurrence of the rock-paper-scissors cycle in various real-world scenarios. Theoretical models of the rock-paper-scissors cycle have been explored in both two-player[5] and multiplayer game frameworks[6], contributing to an understanding of its underlying properties—as a consequence of strategy diversity, the intransitive interaction may emerge spontaneously. This phenomenon can be illustrated when we extend the basic two-strategy model of the evolution of cooperation by adding additional strategies that punish defectors[7,8] or reward cooperators[9,10]. The additional strategies are necessary when

considering more realistic models, which underlines the importance of a multi-strategy approach.

Previous research in evolutionary dynamics primarily focused on two-strategy systems, where the unconditional cooperator and defector strategies represent the fundamental conflict of individual and collective interests[11]. While cooperation can maximize mutual benefits, defection, despite offering higher personal payoff, reduces overall benefits to others. Consequently, defection often appears as the dominant strategy. An escape route from this dilemma could be a spatially structured population[12,13], where individuals interact with fixed neighbors but still adopt the strategies of those with higher payoffs. This setting allows cooperation to form clusters, utilizing the advantage of collective payoffs thus resisting the invasion of defection, a concept known as spatial reciprocity[14]. It is recognized that no simple closed-form solution exists for general evolutionary dynamics in

[1]Department of Computational and Data Sciences, George Mason University, Fairfax, VA 22030, USA. [2]Faculty of Natural Sciences and Mathematics, University of Maribor, Koroška cesta 160, 2000 Maribor, Slovenia. [3]Community Healthcare Center Dr. Adolf Drolc Maribor, Vošnjakova ulica 2, 2000 Maribor, Slovenia. [4]Complexity Science Hub Vienna, Josefstädterstraße 39, 1080 Vienna, Austria. [5]Department of Physics, Kyung Hee University, 26 Kyungheedae-ro, Dongdaemun-gu, Seoul, Republic of Korea. [6]Institute of Technical Physics and Materials Science, Centre for Energy Research, P.O. Box 49, H-1525 Budapest, Hungary. ✉e-mail: CqWang814921147@outlook.com

structured populations, unless by chance P = NP, *Polynomial time* equals to *Nondeterministic Polynomial time*[15]. However, in the weak selection limit, where the influence of the game on strategy updates is marginal, analytical solutions have been obtained from infinite[16] to finite populations[17], and from regular[18,19] to arbitrary graphs[20–23]. This line of research has led to the development of evolutionary graph theory[14].

In evolutionary graph theory, a widely used mathematical technique is the pair approximation[24–28]. This method applies to infinite populations on regular graphs and has revealed the well-known '$b/c > k$' rule, which states that evolution favors cooperation when the benefit-to-cost ratio exceeds the number of neighbors[29]. Pair approximation is also capable of analyzing more complex models, including unequal interaction and dispersal graphs[30], asymmetric networks[31], and stochastic games[32], predicting simulation outcomes with high accuracy. Notably, pair approximation has been applied to multi-strategy two-player games[33], leading to the replicator equations for arbitrary $n$-strategy two-player games on a regular graph, as an important extension of the traditional replicator equations used in well-mixed populations[34].

Unlike two-player games, multiplayer games exhibit much greater complexity, primarily due to their potentially nonlinear payoff functions[35]. In a structured population, multiplayer games require each individual to organize a game within their neighbors and themselves, which implies that individuals participate in games organized by both themselves and their neighbors, thereby interacting with second-order neighbors. Such interactions lead to higher-order interactions[36,37], which cannot be simply reduced to a superposition of pairwise interactions. The complexity of multiplayer games can also be illustrated from the perspective of structure coefficients on graphs: a two-strategy two-player game needs only one structure coefficient[38], a multi-strategy two-player game requires three[39,40], but a two-strategy $(k+1)$-player game needs as many as $k$ structure coefficients[41,42]. The

number of potential equilibrium points in general multiplayer games also indicates their complexity[43,44]. Even so, in the absence of triangle motifs, two-strategy multiplayer games can still be theoretically analyzed using pair approximation[45,46], whose results are consistent with predictions obtained by other more precise methods[47,48].

With two-strategy two-player[29], multi-strategy two-player[33], and two-strategy multiplayer games[46] all thoroughly studied, the analytical solution for multi-strategy multiplayer games on graphs remains unexplored. The range of potential models for multiplayer games with more than two strategies is vast, drawing from co-evolutionary strategies such as punishment[8,49–51], reward[9,52,53], and the loner strategy[4,54]. Multi-strategy systems in multiplayer games have unique characteristics that multistrategy two-player games do not capture: the payoff function can be nonlinear. For example, in pool punishment, the payoff structure depends solely on whether there is at least one punishing player among the $k+1$ players. This uniqueness reinforces the significance of studying multistrategy multiplayer games.

However, previous research on these games on graphs has largely been limited to numerical simulations, which do not allow for the exploration of the complete parameter space. In the absence of mathematical tools for evolutionary graph theory in multi-strategy multiplayer games, recent studies have attempted to bypass this challenge by incorporating the third strategy within the existing two strategies. For instance, punishing or rewarding behaviors have been added to the existing cooperation strategy in the traditional two-strategy system[55,56]. This approach allows for the examination of additional mechanisms like punishment and reward within the two-strategy system's framework. Yet, these alternative attempts still could not capture further rich dynamics, such as cyclic dominance, which is only possible in systems with at least three strategies.

In this work, we provide an analytical framework that addresses the gaps in multi-strategy multiplayer games in the realm of evolutionary graph theory. Inspired by the Balls-and-Boxes problem, we

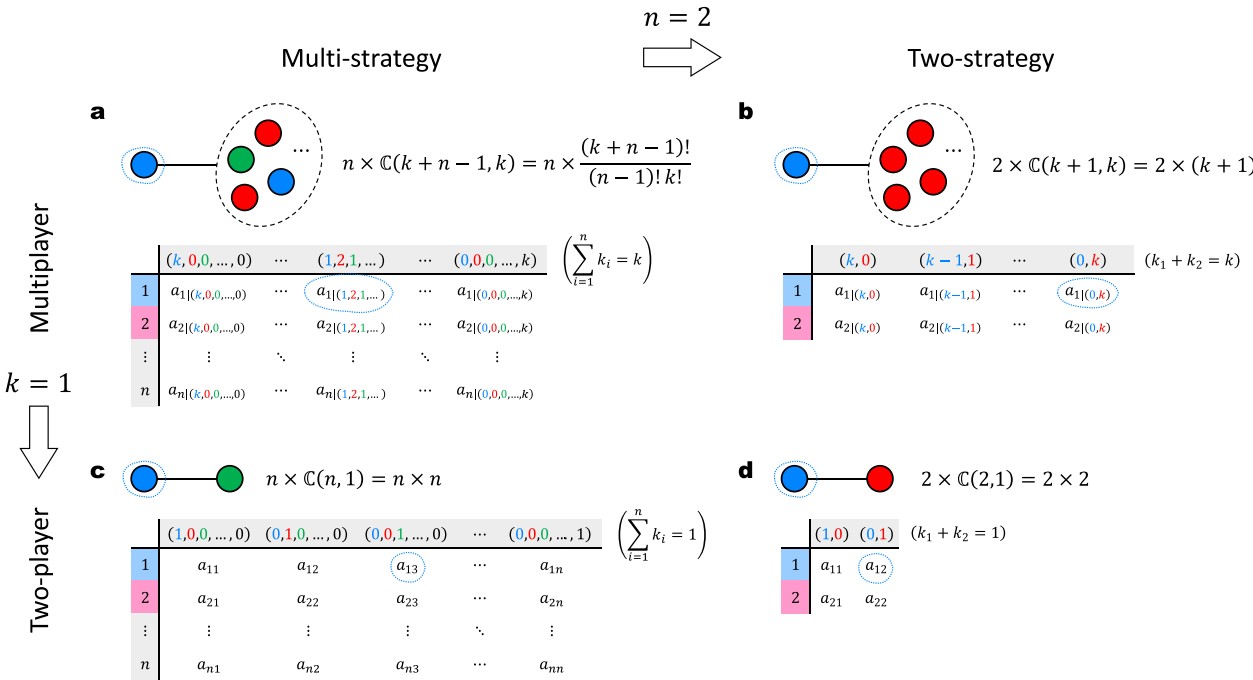

**Fig. 1 | Generalized payoff matrix for multi-strategy multiplayer games, reducible to two-strategy or two-player formats. a** Payoff matrix for $n$-strategy $(k+1)$-player games: with $n$ strategies and $k$ co-players, the matrix size is $n \times \mathbb{C}(k+n-1, k)$. **b** Payoff matrix for 2-strategy $(k+1)$-player games[46]: with $n = 2$ strategies and $k$ co-players, the matrix size is $2 \times \mathbb{C}(k+1, k)$. **c**, Payoff matrix for $n$-strategy 2-player games[33]: with $n$ strategies and $k = 1$ co-players, the matrix size is $n \times \mathbb{C}(n, 1)$. **d**, Payoff matrix for 2-strategy 2-player games[29]: with $n = 2$ strategies and $k = 1$ co-players, the matrix size is $2 \times \mathbb{C}(2, 1)$.

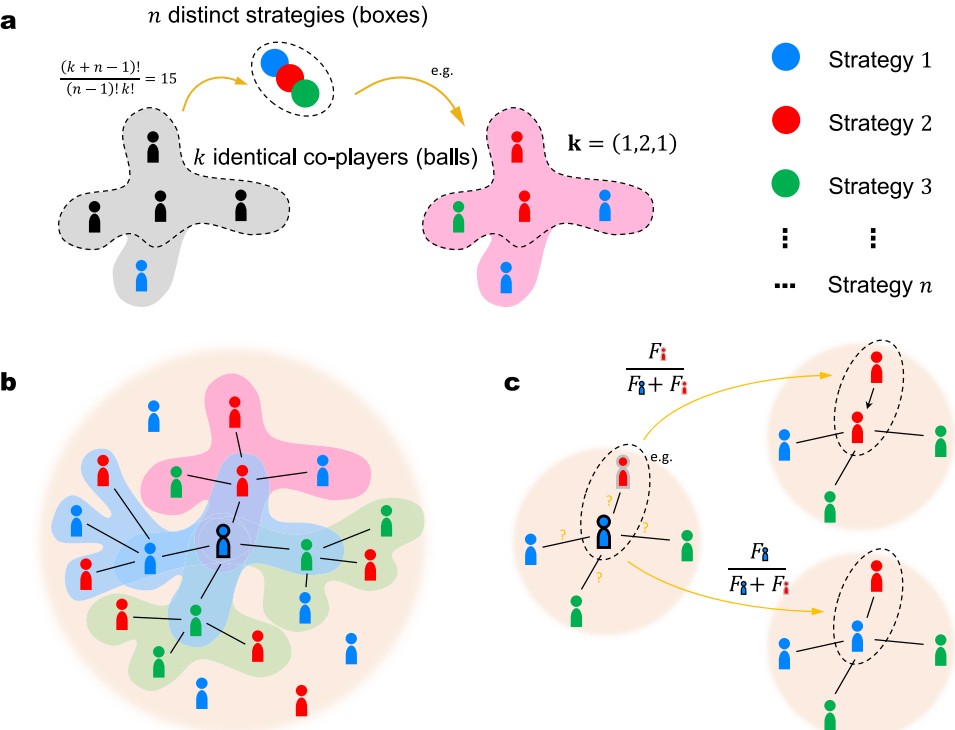

**Fig. 2 | The Balls-and-Boxes problem, payoff calculation, and strategy updates.**
**a** To determine a co-player configuration, we distribute $k$ identical co-players (balls) into $n$ distinct strategies (boxes), corresponding to the classic Balls-and-Boxes problem. For instance, with $k = 4$ and $n = 3$, there are 15 possible co-player configurations. **b** An individual accumulates payoffs from $1 + k$ multiplayer games, organized by itself and its $k$ neighbors. Each multiplayer game involves $k + 1$ players, including the organizer and its neighbors. **c** Strategy updates in an individual are governed by the pairwise comparison rule. A random neighbor is selected, and the focal individual either adopts the neighbor's strategy or maintains its original strategy, based on a probability proportional to the fitness (transformed from payoffs) in the pair.

demonstrate that for a given multi-strategy multiplayer game, counting the co-player configurations of a focal individual is equivalent to distributing $k$ identical co-players into $n$ distinct strategies (Fig. 1). On this basis, we develop a bottom-up approach for calculating the group-based payoff of individuals on regular graphs, deriving replicator equations on regular graphs in the weak selection limit and the absence of triangle motifs. Our results include two commonly used update rules, namely pairwise comparison (PC)[57] and death-birth (DB)[29], applicable to arbitrary multiplayer games with any multi-strategy space in structured populations, where each individual has the same number of neighbors. Using the punishment mechanism[7,58,59] in the context of the tragedy of the commons[60] as an example, we explore the well-known second-order free-riding problem analytically, obtaining an accurate threshold of punishment strength necessary to resolve the social dilemma in structured populations. Additionally, our theoretical solutions can qualitatively reproduce the phase diagrams observed in previous numerical simulation studies under non-marginal selection strength.

## Results

### Model overview

We consider an infinite population on a regular graph, where each individual has $k$ neighbors. An individual can adopt one of $n$ strategies, labeled by the numbers $1, 2, \ldots, n$. On a regular graph, the number of co-players in every multiplayer game is equivalent to the constant number $k$ of neighbors. For a given individual, suppose that there are $k_1$ co-players employing strategy 1, $k_2$ co-players employing strategy 2, and so on, up to $k_n$ co-players employing strategy $n$. In this context, the co-player configuration of an individual can be represented by $\mathbf{k} = (k_1, k_2, \ldots, k_n)$, which satisfies the condition $\sum_{l=1}^{n} k_l = k$. As

illustrated in Fig. 2a, counting the number of possible configurations of $\mathbf{k}$ is analogous to the classic Balls-and-Boxes problem, distributing $k$ identical balls (i.e., co-players) into $n$ distinct boxes (i.e., strategies), allowing for the possibility of empty boxes (e.g., $k_1 = 0$). Hence, there are $\mathbb{C}(k + n - 1, k) = (k + n - 1)!/[(n - 1)!k!]$ possible configurations of co-player strategy configurations $\mathbf{k}$.

Interaction occurs between an individual and its $k$ co-players. In a multiplayer game involving the focal individual and $k$ co-players, the payoff is uniquely determined by the strategy of the focal individual and the strategy configuration of the co-players. For a focal individual employing strategy $i$ with the co-player configuration $\mathbf{k}$, its payoff is denoted by $a_{i|\mathbf{k}}$. It can be observed that the 'generalized payoff matrix' comprises $n \times \mathbb{C}(k + n - 1, k)$ elements represented by $a_{i|\mathbf{k}}$ through all possible focal strategies $i = 1, 2, \ldots, n$ and co-player strategy configurations $\mathbf{k}$. For two-strategy two-player games ($n = 2$, $k = 1$), the number of elements in the payoff matrix reduces to $2 \times \mathbb{C}(2,1) = 4$; for multi-strategy two-player games ($k = 1$), it reduces to $n \times \mathbb{C}(n,1) = n^2$; for two-strategy multiplayer games ($n = 2$), it reduces to $2 \times \mathbb{C}(k + 1, k) = 2(k + 1)$ (Fig. 1).

The accumulated payoff of a focal individual is collected from the $1 + k$ games organized by itself and its neighbors, as depicted in Fig. 2b. Upon obtaining the accumulated payoffs $\pi$, we convert them into fitness, denoted as $F = \exp(\delta\pi)$[21,48,61,62]. Strategies that yield higher fitness are more likely to reproduce. Here, $\delta \to 0^+$ represents a weak selection limit. The rationale behind weak selection is that, in reality, many factors other than the investigated game influence the probability of reproduction[29].

There are various commonly used strategy update rules. For simplicity, we focus on the pairwise comparison (PC) rule[57] in the main text (another well-known rule, the death-birth, is discussed

in Supplementary Information). During each elementary step, an individual $A$ and one of its neighbors $B$ are randomly selected from the population. Their payoffs are computed as $\pi_A$ and $\pi_B$ and then transformed into fitness values $F_A$ and $F_B$. Individual $A$ adopts the strategy of individual $B$ with a probability proportional to their fitness in the pair,

$$W = \frac{F_B}{F_A + F_B} = \frac{1}{1 + \exp[-\delta(\pi_B - \pi_A)]}. \quad (1)$$

Or, individual $A$ keeps its own strategy with the remaining probability $F_A/(F_A + F_B)$. Eq. (1) indicates that individual $A$ has a marginal tendency to either maintain its own strategy or adopt the one of individual $B$, depending on who has higher fitness. The evolution of strategies under the PC update process is illustrated in Fig. 2c.

## Group-based payoff with any number of strategies

To formally analyze the evolutionary dynamics, we construct the system as described in Supplementary Note 1.1. According to pair approximation[29], there are two key concepts, the frequency of $i$-players (i.e., individuals employing strategy $i$), denoted as $x_i$, where $\sum_{i=1}^{n} x_i = 1$, and the probability of an $i$-player being adjacent to a $j$-player, denoted by $q_{i|j}$, with $\sum_{j=1}^{n} q_{j|i} = 1$. By separating different time scales, we find that $q_{i|j} = x_i(k-2)/(k-1) + \theta_{ij}/(k-1)$, where $\theta_{ij} = 1$ if $i = j$ and $\theta_{ij} = 0$ otherwise (Supplementary Note 5.1). In other words, the value of $q_{i|j}$ can be determined by the value of $x_i$.

To express necessary computations, we introduce a variation of $\mathbf{k}$, denoted as $\mathbf{k}_{+l} = (k_1, k_2, ..., k_l+1, ..., k_n)$, where $\sum_{l=1}^{n} k_l = k - 1$. This represents a co-player configuration in which there is at least one $l$-player. Among the remaining $k - 1$ co-players, the numbers of players adopting strategies $1, 2, ..., n$ are $k_1, k_2, ..., k_n$, respectively.

We label the payoff that an individual obtains in a multiplayer game as the single-game payoff. $\langle a_{X|\mathbf{k}} \rangle_Y$ is used to denote the expected single-game payoff for an $X$-player over the possible co-player configurations $\mathbf{k}$, where the $k$ members in $\mathbf{k}$ are neighbors of a $Y$-player, as defined by Eq. (12) in the Methods. Similarly, the notation $\langle a_{X|\mathbf{k}_{+l}} \rangle_Y$ differs in that it is over $k - 1$ unknown members in the possible co-player configurations $\mathbf{k}_{+l}$, with one known $l$-player.

Furthermore, we use the notation $\langle \pi_X^{\mathbf{k}} \rangle$ to represent the expected accumulated payoff of an $X$-player obtained in the $1 + k$ games organized by the player and its neighbors, across all possible neighbor configurations $\mathbf{k}$ of the $X$-player, defined by Eq. (10) in the Methods. Similarly, $\langle \pi_X^{\mathbf{k}_{+i}} \rangle$ denotes the expected accumulated payoff over the configurations where the remaining $k - 1$ neighbors are unknown besides a known $i$-player, as defined by Eq. (11) in the Methods.

Through bottom-up calculations from the microscopic level (Methods), we establish the following relationship between the expected accumulated and single-game payoffs. For $i$-players, the relation is given by

$$\left\langle \pi_i^{\mathbf{k}} \right\rangle = \langle a_{i|\mathbf{k}} \rangle_i + k \sum_{l=1}^{n} q_{l|i} \langle a_{i|\mathbf{k}'_{+l}} \rangle_l. \quad (2)$$

Intuitively, the expected accumulated payoff of $i$-players, $\langle \pi_i^{\mathbf{k}} \rangle$, is composed by the expected single-game payoff from the game they organize, $\langle a_{i|\mathbf{k}} \rangle_i$, and the games organized by their $k$ neighbors, $k \sum_{l=1}^{n} q_{l|i} \langle a_{i|\mathbf{k}'_{+l}} \rangle_l$. Here, the different notation $\mathbf{k}' = (k'_1, k'_2, ..., k'_n)$ from $\mathbf{k}$ is an independent configuration to clarify the priority in the summation.

A further concept is the expected accumulated payoff of a $j$-player who has at least one $i$-player as a neighbor, which is related to the expected single-game payoff as follows:

$$\left\langle \pi_j^{\mathbf{k}_{+i}} \right\rangle = \langle a_{j|\mathbf{k}_{+i}} \rangle_j + \langle a_{j|\mathbf{k}'_{+i}} \rangle_i + (k-1) \sum_{l=1}^{n} q_{l|j} \langle a_{j|\mathbf{k}'_{+l}} \rangle_l. \quad (3)$$

Here, $\langle a_{j|\mathbf{k}_{+i}} \rangle_j$, $\langle a_{j|\mathbf{k}'_{+i}} \rangle_i$, and $(k-1) \sum_{l=1}^{n} q_{l|j} \langle a_{j|\mathbf{k}'_{+l}} \rangle_l$ are the expected single-game payoff from the game organized by the $j$-player itself, the game organized by the fixed $i$-player neighbor, and the games organized by the remaining $k - 1$ neighbors of the $j$-player.

## General replicator equations

The evolution of frequencies $x_1, x_2, ..., x_n$ can be deduced through the microscopic strategy update process. Specifically, in an infinite population, i.e., $N \to \infty$, a single unit of time comprises $N$ elementary steps, ensuring that each individual has an opportunity to update their strategy. During each elementary step, the frequency of $i$-players increases by $1/N$ when a focal $j$-player (where $j \ne i$) is chosen to update its strategy and is replaced by an $i$-player. Similarly, the frequency of $i$-players decreases by $1/N$ when a focal $i$-player is selected to update its strategy and the player who takes the position is not an $i$-player. Based on this perception, we derive a simple form of the replicator equations for $i = 1, 2, ..., n$ in the weak selection limit (Supplementary Note 2):

$$\dot{x}_i = \frac{\delta}{2} x_i \left( \left\langle \pi_i^{\mathbf{k}} \right\rangle - \sum_{j=1}^{n} q_{j|i} \left\langle \pi_j^{\mathbf{k}_{+i}} \right\rangle \right). \quad (4)$$

We find that Eq. (4) offers an intuitive understanding, if we introduce the following two concepts: (1) $\pi_i^{(0)} = \langle \pi_i^{\mathbf{k}} \rangle$, the expected accumulated payoff of the $i$-player (zero steps away on the graph), and (2) $\pi_i^{(1)} = \sum_{j=1}^{n} q_{j|i} \langle \pi_j^{\mathbf{k}_{+i}} \rangle$, the expected accumulated payoff of the $i$-player's neighbors (one step away on the graph). These concepts suggest that $\dot{x}_i \propto x_i(\pi_i^{(0)} - \pi_i^{(1)})$. Under pairwise comparison, the reproduction rate of $i$-players is dependent on how their accumulated payoff exceeds that of their neighbors. In essence, the evolution of $x_i$ is the competition between an individual and its first-order neighbors, which aligns with the results obtained by a different theoretical framework in two-strategy systems[18,20,32]. We further extend it to $n$-strategy systems in the framework of pair approximation. We also verify that the death-birth rule is essentially the competition between an individual and its second-order neighbors for $n$-strategy systems (Supplementary Information).

Applying Eqs. (2) and (3) to Eq. (4), we can transform the expected accumulated payoff in the replicator equations into the expected single-game payoff, as shown in Eq. (13) in the Methods, which keeps the simplest irreducible computational complexity given the payoff structure $a_{i|\mathbf{k}}$. In particular, we only need to calculate two types of quantities, $\langle a_{i|\mathbf{k}_{+j}} \rangle_i$ and $\langle a_{i|\mathbf{k}_{+j}} \rangle_j$ for $i, j = 1, 2, ..., n$, based on the given payoff structure $a_{i|\mathbf{k}}$. The diagonal elements of these quantities coincide, as demonstrated in Eqs. (14) and (15) in the Methods. Therefore, for any given payoff structure $a_{i|\mathbf{k}}$, there are at most $(2n-1)n$ distinct quantities to calculate manually when determining the replicator equations. The computational complexity is thus $O(n^2)$, square of the number of strategies, which can be solved within polynomial time. We also find that the computational complexity under the death-birth rule is $O(n^3)$, cubic of the number of strategies, which can also be solved within polynomial time (Supplementary Information).

For specific payoff structures, the computational complexity can be further reduced. A common example is linear systems. In such systems, the payoff function includes at most linear terms in $k_1, k_2, ..., k_n$. This allows us to express the general payoff function as $a_{i|\mathbf{k}} = \sum_{j=1}^{n} b_{ij} k_j + c_i$, where $b_{ij}$ represents the coefficient of the linear term and $c_i$ is the constant term for $i, j = 1, 2, ..., n$. Applying this special payoff structure to Eq. (13) in the Methods, we can obtain a simplified form of the replicator equation for linear systems,

$$\dot{x}_i = \frac{\delta(k-2)}{2(k-1)} x_i \left( (k+1)(\bar{\pi}_i - \bar{\pi}) + 3 \sum_{j=1}^{n} x_j(b_{ii} - b_{ij} - b_{ji} - b_{jj}) + 6 \sum_{j=1}^{n} \sum_{l=1}^{n} x_j x_l b_{jl} \right). \quad (5)$$

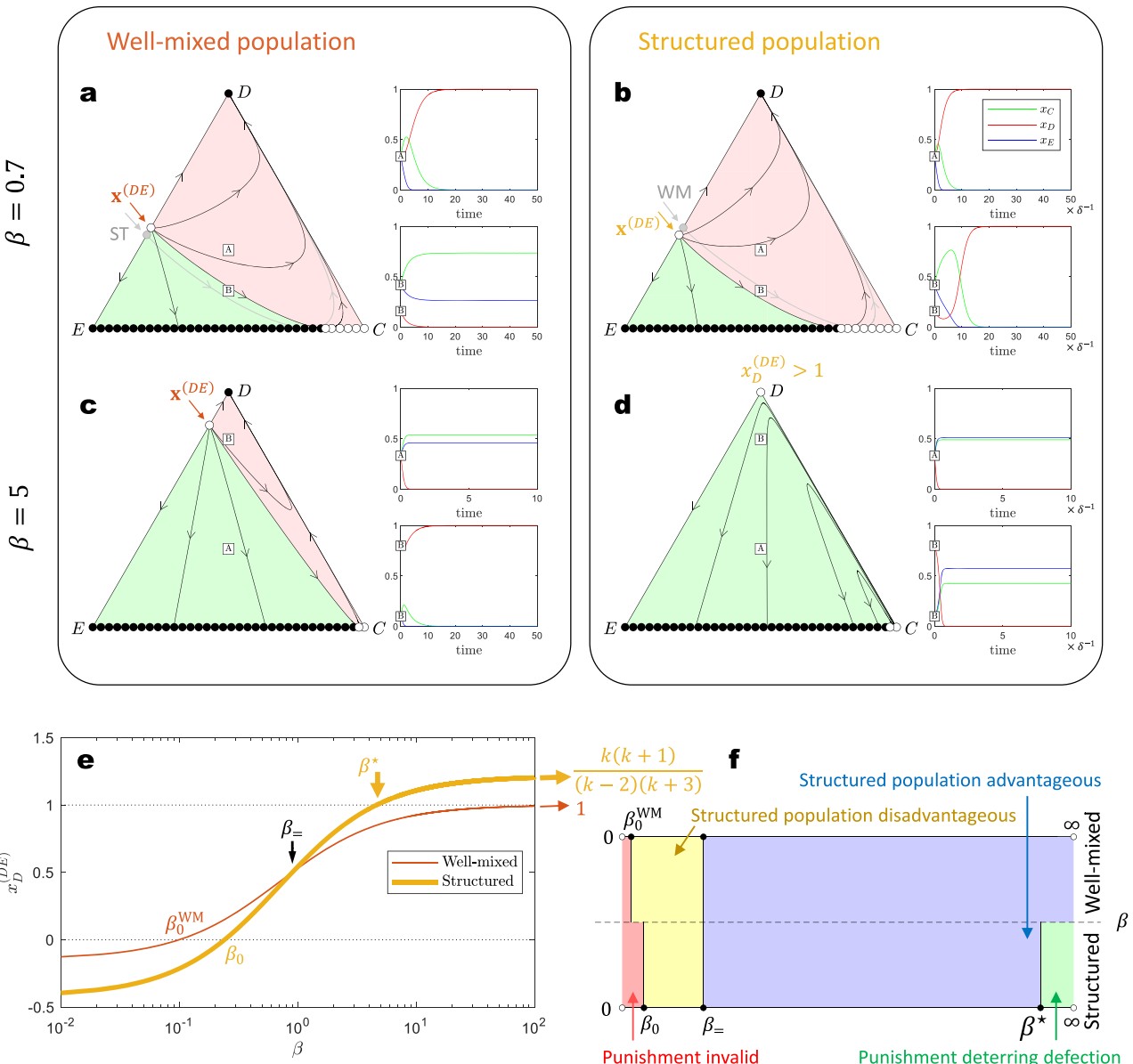

**Fig. 3 | Peer punishment can resolve the social dilemma of public goods game in structured populations.** The traditional replicator dynamics produce results for well-mixed populations, and our framework allows for exploring the dynamics in structured populations. **a** and **b** The state space is bifurcated by $\mathbf{x}^{(DE)}$ and $\mathbf{x}_*^{(CE)}$. The final state, either $D$ or $(C+E)_V$, is determined by the initial conditions. Under mild punishment ($\beta = 0.7$), a structured population hinders cooperation by reducing the state space leading to the $(C+E)_V$ outcome. **c** and **d** Conversely, with strong punishment ($\beta = 5$), structured populations consistently result in the extinction of defection, thereby resolving the social dilemma in public goods games. In contrast,

the state space in well-mixed populations remains divided into two distinct basins. **e** As the punishment strength $\beta$ increases, $x_D^{(DE)}$ increases, expanding the initial space leading to the $(C+E)_V$ outcome. In well-mixed populations, $x_D^{(DE)} \to 1$ as $\beta \to \infty$, and the basin leading to defection cannot be completely eliminated. However, in structured populations, $x_D^{(DE)} \to k(k+1)/[(k-2)(k+3)] > 1$ when $\beta > \beta^*$, invariably resulting in the extinction of defection. **f** The diagram of the different effects of punishment in well-mixed versus structured populations. Structured populations are advantageous in promoting cooperation under strong punishment but are less effective when the punishment is mild. **Input parameters**: $r = 3$, $c = 1$, $\alpha = 0.7$, $k = 4$.

Here, $\bar{\pi}_i$ and $\bar{\pi}$ denote the mean payoff of $i$-players and all players in a well-mixed population, which can be directly calculated using the traditional replicator dynamics approach (Methods).

As a frequently studied example, the public goods game involves $n = 2$ strategies within a linear payoff structure. Strategy 1, cooperation ($C$), pays a cost $c$ which is multiplied by a synergy factor $r$ and distributed among all $k + 1$ players, while strategy 2, defection ($D$), pays nothing. The payoff structure can be expressed as $b_{11} = b_{21} = rc/(k+1)$, $b_{12} = b_{22} = 0$, $c_1 = rc/(k+1) - c$, $c_2 = 0$. Consequently, $\dot{x}_i \propto x_i(\bar{\pi}_i - \bar{\pi})$, indicating that evolution favors cooperation when $r > k + 1$ (Supplementary Note 3.1). Coincidentally, the public

goods game exhibits an equivalence between well-mixed and structured populations under pairwise comparison[63], a phenomenon not necessarily observed under other update rules[45]. This equivalence provides a unique opportunity: when introducing additional strategies into the public goods game, the distinct effects of these new strategies in structured populations can be isolated without interference from the existing two strategies. For a general condition when pairwise comparison equates well-mixed and structured populations, we refer to the Supplementary Note 3.1.3.

We apply the multi-strategy multiplayer framework to various additional mechanisms in public goods games, including

punishment[8,51] ($n = 3$), reward[64] ($n = 3$), and multi-stage investment[65] ($n = 4$) (Supplementary Note 3). Here, we present the applications to two punishment types, peer and pool punishment, by which we revisit the well-known second-order free-rider problem in structured populations.

## Peer punishment in public goods games

In public goods games with peer punishment[66,67], the payoff structure is linear (Supplementary Note 3.2), which allows us to utilize Eq. (5) directly.

There are $n = 3$ strategies: 1 = Cooperation ($C$), 2 = Defection ($D$), and 3 = Peer punishment ($E$). Besides the two strategies in the public goods game, the third strategy, peer punishment, pays a cost $\alpha$ for punishing a co-player who defects. A defector, when punished, incurs a fine $\beta$. Thus, given $k_2$ defective co-players, a punishing player has $\alpha k_2$ paid, and given $k_3$ punishment co-players, a defector has $\beta k_3$ charged. Furthermore, it is assumed that punishing players also perform the cooperative behavior, investing $c$ to the common pool. This makes the strategy $C$ the second-order free-rider who exploits the effort in punishment of strategy $E$.

The first question is how the behaviors of peer punishment in structured populations, obtained by our framework (Supplementary Note 3.2), differ from the ones in a well-mixed population. We find that peer punishment introduces a bi-stable space of the system state, as seen in Fig. 3a, b. Even when $r < k + 1$, the system can either evolve to a final state where strategies $E$ and $C$ coexist, or to a state dominated by strategy $D$, depending on the initial conditions. As the punishing fine $\beta$ increases, the basin of attraction for strategy $D$ diminishes. In a well-mixed population, strategy $D$ maintains a basin of attraction regardless of the punishment strength (Fig. 3c). This aligns with previous findings that peer punishment does not truly resolve social dilemmas in well-mixed populations[59]. However, in structured populations, we observe that the basin of attraction for strategy $D$ can be entirely eliminated if the punishing fine $\beta$ exceeds a critical threshold, $\beta > \beta^\star$, where

$$\beta^\star = \frac{k+1}{3}\left(-\frac{rc}{k+1} + c + k\alpha\right) - \alpha. \tag{6}$$

Consequently, in such scenarios, the system consistently converges to a coexistence of strategies $E$ and $C$ (Fig. 3d). The numerical observation from previous research suggests that peer punishment can effectively resolve social dilemmas in structured populations. Our analysis adds an analytical perspective to this conclusion.

The distinct roles of peer punishment in well-mixed and structured populations can be attributed to the fraction of defectors, $x_D^{(DE)}$, in an unstable edge equilibrium, $\mathbf{x}^{(DE)} = (0, x_D^{(DE)}, 1 - x_D^{(DE)})$, as presented in Fig. 3e. When $x_D^{(DE)} > 1$, this unstable equilibrium disappears, rendering the $D$-vertex equilibrium unstable. In a well-mixed population, $x_D^{(DE)} < 1$ and $x_D^{(DE)} \to 1$ as $\beta \to \infty$, indicating that the described scenario is unattainable. However, in structured populations, $x_D^{(DE)} > 1$ becomes feasible once $\beta > \beta^\star$. Additionally, peer punishment acts as a double-edged sword. When $x_D^{(DE)} < 0$, the system invariably converges to the full defection state, signifying ineffective punishment. As the punishing fine $\beta$ increases, peer punishment first becomes effective in well-mixed populations when $\beta > \beta_0^{WM}$. Structured populations, in contrast, require a higher $\beta_0$ value for punishment to be effective. In particular, peer punishment is less advantageous in structured populations than in well-mixed populations when $\beta < \beta_=$ (Fig. 3a, b). However, structured populations gain an advantage when $\beta > \beta_=$, and can eventually lead to the extinction of defection at sufficient high $\beta > \beta^\star$ values. The comparison between well-mixed and structured populations in relation to the punishing fine is illustrated in Fig. 3f, and the expressions for key $\beta$ values are listed in Fig. 4.

We also compare the analytical predictions by our framework to the results from previous work, which was only at a numerical level. As shown in Fig. 5, we find our analytical results align qualitatively with the $\alpha$-$\beta$ phase diagrams presented in previous research[51]. Although there are differences in detail between the results obtained from non-marginal selection through numerical simulations (Fig. 5a, c) and those derived under weak selection via analytical solutions (Fig. 5b, d), both approaches consistently predict unique behaviors in structured populations that are absent in well-mixed populations. For instance, both the nonmarginal and weak selection strengths indicate the existence of a $(C + E)_V$ phase at low $\alpha$ and high $\beta$, where strategy $D$ becomes extinct and strategies $C$ and $E$ coexist, equivalent to the Voter model[68,69]. Moreover, at moderate levels of $\alpha$ and $\beta$, we anticipate a $D \Leftrightarrow (C+E)_V$ phase under weak selection. In this phase, the system evolves towards either $D$ or $(C+E)_V$ based on the initial state, although strategy $C$ may eventually become extinct due to the continuous introduction of a small number of defectors[70]. A similar phase, named $D_{h(E)}$, is detected under non-marginal selection. The term '$h$' denotes 'homoclinic instability', implying that strategy $E$ can overcome $D$ through a nucleation mechanism, particularly if a small colony of $E$ players survives after the extinction of cooperators. This likelihood increases with larger populations.

## Pool punishment in public goods games

Another example is pool punishment in public goods games[8,71]. From the perspective of computational complexity, pool punishment differs from peer punishment in its nonlinear payoff structure, which requires utilizing Eq. (13).

Similarly, there are $n = 3$ strategies: 1 = Cooperation ($C$), 2 = Defection ($D$), and 3 = Pool punishment ($O$). Again, based on the 2-strategy public goods game, the third strategy, pool punishment, contributes a cost $\alpha$ to the public pool for punishment. A defector is punished with a fine $\beta$ if the public pool for punishment has funds (i.e., there is at least one punisher among the co-players); if no punishers are present, the defector incurs no charge. Irrespective of the number of defecting co-players $0 \le k_2 \le k$, a punishing player pays $\alpha$. It is also assumed that those employing pool punishment engage in cooperative behavior, investing $c$ to the common pool, making the strategy $C$ a second-order free-rider.

Our analysis in structured populations (Supplementary Note 3.3) and the traditional analysis for well-mixed populations reveal that pool punishment does not change the fact that the system cannot converge to a defection-free state when $r < k + 1$, as demonstrated in Fig. 6a, b. However, along the $DO$ edge (i.e., without the presence of strategy $C$), the system can evolve to a final state of either full $D$ or full $O$, depending on the initial conditions. As the punishing fine $\beta$ increases, the attraction basin for strategy $D$ shrinks. In well-mixed populations, strategy $D$ retains an attraction basin regardless of the punishment strength (Fig. 6c). This is consistent with previous findings that pool punishment does not effectively resolve social dilemmas in well-mixed populations[59]. However, in structured populations, the attraction basin for strategy $D$ can be completely eliminated if the punishing fine $\beta$ exceeds a critical threshold, $\beta > \beta^\star$, where

$$\beta^\star = \frac{k+1}{2}\left(-\frac{rc}{k+1} + c + \alpha\right). \tag{7}$$

Given that $O$ and $C$ are still unstable, the system consequently enters a cyclic dominance pattern among $D$, $O$, and $C$ in such scenarios (Fig. 6d). The cyclic dominance follows the sequence $D \to O \to C \to D \to \cdots$. Numerical observations from previous studies suggest that pool punishment can resolve social dilemmas in structured populations by inducing a cycle of defection[8]. Our theoretical approach provides accurate insight into this phenomenon.

## Peer punishment

| Equilibrium of $(x_C, x_D, x_E)$ | Well-mixed population | Structured population | Note |
|---|---|---|---|
| $(0,1,0)$ | Always stable | Unstable when $\beta > \beta^\star$ | Peer punishment can deter defection in structured populations |
| $\left(0, x_D^{(DE)}, 1 - x_D^{(DE)}\right)$ | Exists when $\beta > \beta_0^{WM}$ <br> Unstable | Exists when $\beta_0 < \beta < \beta^\star$ <br> Unstable | $\beta < \beta_=$: Well-mixed population is more advantageous <br><br> $\beta > \beta_=$: Structured population is more advantageous |
| $\left(x_C^{(CE)}, 0, 1 - x_C^{(CE)}\right)$ | $0 \le x_C^{(CE)} \le 1$ <br> Stable at $x_C^{(CE)} < x_{C,\star}^{(CE)}$ | $0 \le x_C^{(CE)} \le 1$ <br> Stable at $x_C^{(CE)} < x_{C,\star}^{(CE)}$ | Another demarcation of the attraction basins |

$$\beta^\star = \frac{k+1}{3}\left(-\frac{rc}{k+1} + c + k\alpha\right) - \alpha$$

| | | | |
|---|---|---|---|
| $x_D^{(DE)} = \frac{1}{k(\alpha+\beta)}\left(\frac{rc}{k+1} - c + k\beta\right)$ | $x_D^{(DE)} = \frac{1}{(k-2)(k+3)(\alpha+\beta)}\left[(k+1)\left(\frac{rc}{k+1} - c + k\beta\right) - 3(\alpha+\beta)\right]$ | $\beta_= = \frac{2}{k}\left(-\frac{rc}{k+1} + c\right) + \alpha$ | |
| $\beta_0^{WM} = \frac{1}{k}\left(-\frac{rc}{k+1} + c\right)$ | $\beta_0 = \frac{k+1}{k^2+k-3}\left(-\frac{rc}{k+1} + c\right) + \frac{3\alpha}{k^2+k-3}$ | | |
| $x_{C,\star}^{(CE)} = 1 + \left(\frac{rc}{k+1} - c\right)\frac{1}{k\beta}$ | $x_{C,\star}^{(CE)} = 1 + \left(\frac{rc}{k+1} - c\right)\frac{1}{k\beta - \frac{3(\alpha+\beta)}{k+1}}$ | | |

**Fig. 4 | Analytical results of public goods game with peer punishment in both well-mixed and structured populations.** This table summarizes the equilibrium points and their stability by analyzing the peer punishment system. The analytical forms of all key concepts are presented. See Supplementary Note 3.2 for details of the analysis.

Similarly, the distinct impacts of pool punishment in well-mixed and structured populations can be identified by the fraction of defectors, $x_D^{(DO)}$, in an unstable edge equilibrium, $\mathbf{x}^{(DO)} = (0, x_D^{(DO)}, 1 - x_D^{(DO)})$, as shown in Fig. 6e. When $x_D^{(DO)} > 1$, this unstable equilibrium vanishes, leading to instability of the $D$-vertex equilibrium. In well-mixed populations, $x_D^{(DO)} < 1$ and $x_D^{(DO)} \to 1$ as $\beta \to \infty$, suggesting that the described scenario is unfeasible. Conversely, in structured populations, $x_D^{(DO)} > 1$ becomes true once $\beta > \beta^\star$. Pool punishment also presents a paradoxical effect: when $x_D^{(DO)} < 0$, the system consistently converges to the full defection state, even along the $DO$ edge, indicating ineffective punishment. As the punishing fine $\beta$ increases, pool punishment first becomes effective in well-mixed populations at $\beta > \beta_0^{WM}$. Structured populations, in contrast, require a bit higher $\beta_0$ threshold for effective punishment. Pool punishment is less advantageous in structured populations than in well-mixed populations when $\beta < \beta_=$ (Fig. 6a, b). Nevertheless, structured populations gain an advantage when $\beta > \beta_=$, and can eventually prevent the fixation of defection by inducing cyclic dominance among the three strategies at sufficient high $\beta > \beta^\star$. The comparison between well-mixed and structured populations in relation to the punishing fine is shown in Fig. 6f, with the expressions for key $\beta$ values listed in Fig. 7.

Again, we compare our analytical predictions to the results from previous numerical work. The analytical results are in qualitative agreement with the $\alpha$-$\beta$ phase diagrams from previous research[8], as shown in Fig. 8. Again, while there are detailed differences between outcomes derived from non-marginal selection through numerical simulation (Fig. 8a, c) and those obtained under weak selection with analytical methods (Fig. 8b, d), both approaches indicate distinct behavioral patterns in structured populations that are not observed in well-mixed populations. For example, both non-marginal and weak selection indicate the existence of a cyclic dominance phase,

$(D + C + O)_C$, at low $\alpha$ and high $\beta$, where strategy $D$ invades $C$, strategy $C$ invades $O$, and strategy $O$ invades $D$. Moreover, at moderate levels of $\alpha$ and $\beta$, we predict a $D_{O\rightleftharpoons D}$ phase under weak selection. In this phase, the system consistently evolves towards full $D$ in the three-strategy space; however, in the absence of strategy $C$, the system instead evolves towards either full $O$ or full $D$ based on the initial state. A comparable phase, named $F_{O\rightleftharpoons D}$, is detected under non-marginal selection. The term '$F$' denotes 'fixation', which means that system evolves towards either full $O$ or full $D$.

## Discussion

Spatial evolutionary dynamics under weak selection can be considered as the incorporation of a marginal game effect ($\delta \to 0^+$) on the Voter model[68,69] ($\delta = 0$). In structured populations, identical strategies naturally become adjacent to each other, forming clusters through neutral drift, a process independent of the game, as described by the first-order Taylor expansion in edge dynamics. This inherent tendency for the same strategies to cluster together leads to what is known as spatial reciprocity, a phenomenon captured by the second-order Taylor expansion. Simply put, under weak selection, clusters of the same strategy, caused by spatial structures, unilaterally affect the emergence of cooperation. Conversely, the evolution of cooperation does not influence the spatial pattern of these clusters. This character under weak selection reduces computational complexity, making the closed solution for various evolutionary dynamics such as multi-strategy systems on graphs possible.

In the family of evolutionary graph theory with weak selection and pair approximation, which covers two-strategy two-player, multi-strategy two-player, and two-strategy multiplayer games, we fill in the last piece of the puzzle: the multi-strategy multiplayer games. For a focal individual, we illustrate every possible configuration in which $k$ identical co-players are distributed among $n$ distinct strategies. On this

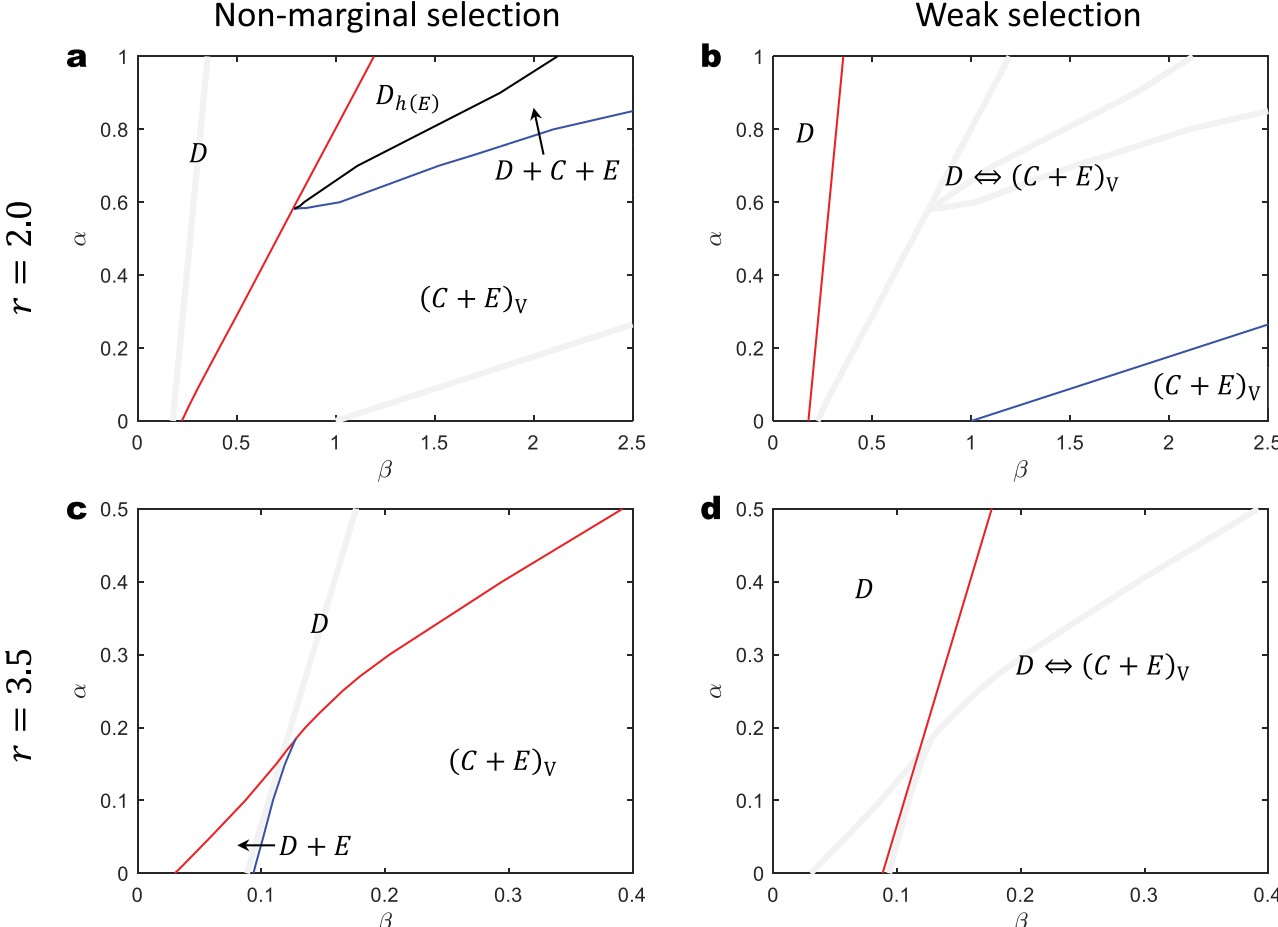

**Fig. 5 | Phase diagrams of the system behavior with respect to punishing cost $\alpha$ and fine $\beta$ are qualitatively similar under non-marginal and weak selection strength. a** and **c** (the data are in agreement with those published in Figs. 3 and 4 from ref. 51), Numerical simulations under non-marginal selection ($\delta = 2$). Here, the different phases are defined as follows: $D$–only $D$ exists; $D_{h(E)}$–only $D$ or $E$ exists based on certain probabilities; $D + C + E$–$D$, $C$, and $E$ coexist; $(C+E)_V$–$C$ and $E$ coexist like in the Voter model. **b** and **d**, The phase diagram created by analytical $\beta_0$ and $\beta^*$ under weak selection ($\delta \to 0^+$), where $\beta_0$ divides the $D$ and $D \Leftrightarrow (C+E)_V$ phases, $\beta^*$ separates the $D \Leftrightarrow (C+E)_V$ and $(C+E)_V$ phases. Specifically, in **b**, $\beta_0 = 3/17 + (3/17)\alpha$ (red), $\beta^* = 1 + (17/3)\alpha$ (blue); in **d**, $\beta_0 = 3/34 + (3/17)\alpha$. The definition of the $D \Leftrightarrow (C+E)_V$ phase–the system finally evolves to either the $D$ phase or the $(C+E)_V$ phase, depending on initial conditions. **Other parameters:** $c = 1$, $k = 4$.

basis, we calculate the group-based payoff for any number of strategies via a bottom-up approach. While we identify each co-player by pair approximation, the $(k + 1)$-player game is treated as a whole and the smallest indivisible unit in our statistical analysis. In this way, the payoff computation for the focal individual is not merely a sum of pairwise interactions, but rather an $n$-element function of the configuration $\mathbf{k} = (k_1, k_2, ..., k_n)$, determined by all co-players simultaneously. The nonlinearity of payoff functions cannot be derived from the superposition pairwise interactions, which reflects the higher-order properties of multi-strategy multiplayer games that are different from multi-strategy two-player games[35].

Building on the group-based payoff calculation, we develop strategy update dynamics on regular graphs using the standard pair approximation method[29,33] under two common update rules: pairwise comparison and death-birth. Interestingly, our general findings are in line with those previously obtained through a different theoretical approach for two-strategy systems[18]. In particular, our $n$-strategy replicator equations imply that pairwise comparison equates to competition among all $n$ strategies between first-order neighbors, while death-birth is equivalent to competition among second-order neighbors. While this is consistent with the previous conclusions for two-strategy systems[18], our results further extend them to the generalized $n$-strategy space.

It is worth mentioning that by contrasting a profile of our results with the other approach in two-strategy public goods games[47,48], we can see the limitations of pair approximation: unlike their approach, which can account for triangle motifs, our pair approximation cannot. According to previous works on pair approximation[45,46], we see that under the death-birth rule (i.e., second-order neighbor competition), pair approximation results align with the other approach only in the absence of triangle motifs. Under the pairwise comparison rule (i.e., first-order neighbor competition), however, triangle motifs appear to have no effect on multiplayer games[48], where the results of pair approximation always match those of the other approach, which considers triangle motifs. This is one reason why pairwise comparison is the primary focus of this paper. For rigor, applying our framework to a specific network structure is best followed by our basic assumption: the absence of triangle motifs. We look forward to a new theory in the future that will cancel this assumption.

For any multi-strategy multiplayer game in our framework, we need only input the payoff function $a_{i|\mathbf{k}}$ for each strategy $i$ across all $(k + n - 1)!/[(n - 1)!k!]$ co-player strategy configurations $\mathbf{k}$. Then, we can apply the general formula provided in this work to obtain the replicator equations on a regular graph. For general payoff functions, we have decomposed the general replicator equation into sums of expected single-game payoffs, as shown in Eq. (13) (for PC updates)

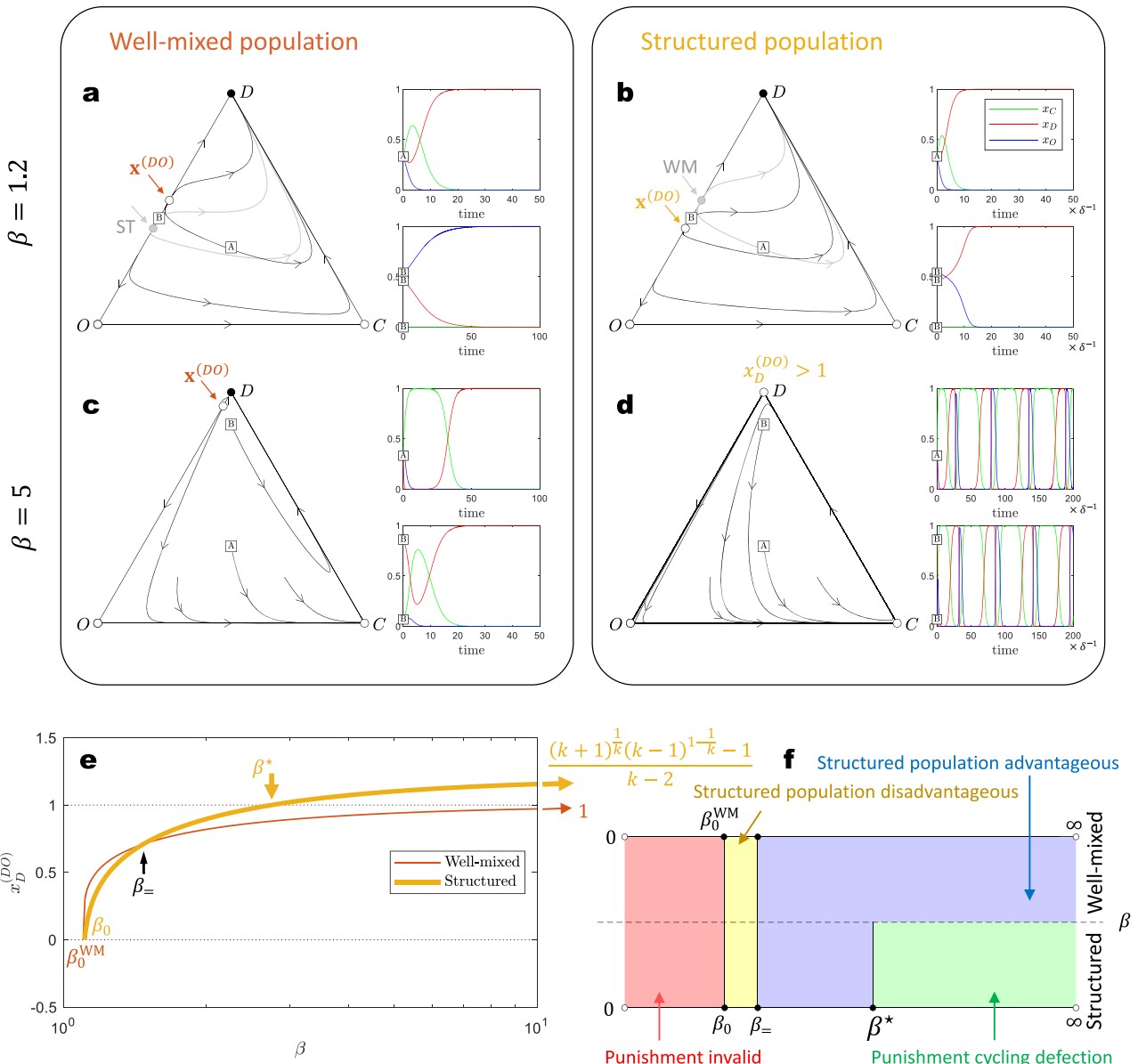

**Fig. 6 | Pool punishment can resolve the social dilemma of public goods game in structured populations. a** and **b** In the three-strategy system space, the state consistently converges to full $D$. However, along the $DO$ edge, an unstable equilibrium point, $\mathbf{x}^{(DO)}$, creates a bi-stable space. In the $D$ versus $O$ dynamics, the final state, either $D$ or $O$, is determined by the initial conditions. Under mild punishment ($\beta = 1.2$), a structured population tends to favor defection, reducing the basin leading to the $O$ outcome. **c** and **d** Conversely, with strong punishment ($\beta = 5$), structured populations result in the cyclic dominance of the three strategies, thereby preventing the full $D$ state in public goods games. In contrast, the state space in well-mixed populations remains two distinct basins on the $DO$ edge,

preventing cyclic dominance. **e** As the punishment strength $\beta$ increases, $x_D^{(DO)}$ increases, expanding the initial space leading to the $O$ outcome. In well-mixed populations, $x_D^{(DO)} \to 1$ as $\beta \to \infty$, and the basin leading to defection cannot be completely eliminated. However, in structured populations, $x_D^{(DO)} \to [(k+1)^{1/k}(k-1)^{1-1/k} - 1]/(k-2) > 1$ when $\beta > \beta^\star$, invariably resulting in the cyclic dominance of the three strategies. **f** The diagram of the different effects of punishment in well-mixed versus structured populations. Structured populations are advantageous in promoting cooperation under strong punishment but are a bit less effective when the punishment is mild. **Input parameters**: $r = 3$, $c = 1$, $\alpha = 0.7$, $k = 4$.

and Supplementary Eq. (S184) (for DB updates). From there, it simplifies the problem to calculating the single games under different strategy configurations and then summing them up. The computation is feasible in polynomial time, which is related to the number of strategies $n$. We find the computational complexity is O($n^2$) for pairwise comparison and O($n^3$) for death-birth. For certain specific payoff functions, the general formula may be further simplified, depending on whether the expected payoff across different strategy configurations has a simple primitive functional form. As an example, we provide a simple general formula for linear payoff functions in both

pairwise comparison and death-birth updates, as shown in Eq. (5) and Supplementary Eq. (S187).

As an application of our theoretical framework, we revisit the second-order free-riding problem. In a simple three-strategy system of cooperation, defection, and cooperative punishment, the defection strategy is a free-rider from cooperation, while the original cooperation is also a free-rider from cooperative punishment. Prior research has shown that costly punishment in well-mixed populations cannot truly resolve social dilemmas[59], although in structured populations it can[8,51]. We further interpret the conclusion within our analytical

## Pool punishment

| Equilibrium of $(x_C, x_D, x_O)$ | Well-mixed population | Structured population | Note | |
|---|---|---|---|---|
| $(0,1,0)$ | Always stable | Saddle point when $\beta > \beta^\star$ $D \to O \to C \to D \to \cdots$ | Pool punishment can cycle defection in structured populations | |
| $\left(0, x_D^{(DO)}, 1 - x_D^{(DO)}\right)$ | Exists when $\beta > \beta_0^{WM}$ Unstable $DO \to O$ or $DO \to D$ | Exists when $\beta_0 < \beta < \beta^\star$ Unstable $DO \to O$ or $DO \to D$ | $\beta < \beta_=$: Well-mixed population is more advantageous | |
| | | | $\beta > \beta_=$: Structured population is more advantageous | |
| $(1,0,0)$ | Saddle point $O \to C \to D$ | Saddle point $O \to C \to D$ | Parts of cyclic dominance | |
| $(0,0,1)$ | Saddle point $DO \to O \to C$ | Saddle point $DO \to O \to C$ | | |

$$\beta^\star = \frac{k+1}{2}\left(-\frac{rc}{k+1} + c + \alpha\right)$$

$$x_D^{(DO)} = \sqrt[k]{1 + \frac{1}{\beta}\left(\frac{rc}{k+1} - c - \alpha\right)}$$

$$\beta_0^{WM} = -\frac{rc}{k+1} + c + \alpha$$

$$x_D^{(DO)} = \frac{k-1}{k-2}\left(-\frac{1}{k-1} + \sqrt[k]{\frac{k+1}{k-1}\left[1 + \frac{1}{\beta}\left(\frac{rc}{k+1} - c - \alpha\right)\right]}\right)$$

$$\beta_0 = \frac{(k+1)(k-1)^{k-1}}{1-(k+1)(k-1)^{k-1}}\left(\frac{rc}{k+1} - c - \alpha\right)$$

$$\beta_= = \frac{\left[(k+1)^{\frac{1}{k}}(k-1)^{1-\frac{1}{k}} - k + 2\right]^k}{1 - \left[(k+1)^{\frac{1}{k}}(k-1)^{1-\frac{1}{k}} - k + 2\right]^k}\left(\frac{rc}{k+1} - c - \alpha\right)$$

**Fig. 7 | Analytical results of public goods game with pool punishment in both well-mixed and structured populations.** This table summarizes the equilibrium points and their stability by analyzing the pool punishment system. The analytical forms of all key concepts are presented. See Supplementary Note 3.3 for details of the analysis.

framework, revealing an accurate threshold for punishment strength $\beta^\star$ in both linear peer punishment and nonlinear pool punishment systems. When the punishment strength $\beta > \beta^\star$, costly punishment can resolve the social dilemma in structured populations. In peer punishment, a sufficiently strong punishment eliminates the attraction basin of full defection in the bi-stable state space. In pool punishment, a strong enough punishment leads the system to a rock-paper-scissors-like cyclic dominance. The results obtained under weak selection also qualitatively reproduce the phase diagrams found in earlier numerical studies under non-marginal selection[8,51], identifying unique phases observable only in structured populations.

In addition, our general $n$-strategy dynamics framework can reduce to classic two-strategy multiplayer game dynamics at $n = 2$. First, although some prior work has explored specific models under pairwise comparison[55,72], to our knowledge, no work has provided a general replicator equation and discussion for two-strategy multiplayer games under pairwise comparison. As a complement to this, we discuss the general replicator equation when $n = 2$ for two-strategy multiplayer games under pairwise comparison in Supplementary Note 2.5. Second, the general replicator equations for two-strategy multiplayer games under death-birth have been discussed by Li et al.[46]. We show that our results obtained under death-birth are identical to theirs at $n = 2$ (Supplementary Note 4.5).

Our theoretical framework is widely applicable, yielding analytical solutions for numerous multistrategy multiplayer game models previously proposed. Besides the two punishment mechanisms investigated in the main text, we also explore the reward mechanism[9,52,53,64] (a mirror mechanism to punishment) and multi-stage public goods game[65] (a four-strategy system) in Supplementary Information. Classic three-strategy games remaining unexplored include tax-based reward and punishment systems[10], the loner strategy[4,54], and so on. Moreover, a pair of additional strategies can be introduced together to create four-strategy systems, such as the competition between peer and pool punishment[51]. In fact, provided coevolutionary factors expressed as payoff functions of co-player configurations, any multi-strategy multiplayer game system can be analyzed within our framework.

## Methods

### Bottom-up statistical quantities

Here, we provide the microscopic details behind the expected payoffs. There are several necessary variations of $\mathbf{k}$ for expressing the details. One is $\mathbf{k}_{+l}$, which still contains $k$ co-players but describe a configuration with at least one $l$-player. The variables satisfy $\sum_{\ell=1}^{n} k_\ell = k - 1$, with the number of $l$-players (when $\ell = l$) written as $k_l + 1$. Similarly, $\mathbf{k}_{-i,+j}$ can describe a configuration where the variables satisfy $\sum_{\ell=1}^{n} k_\ell = k$, with the number of $i$-players as $k_i - 1$ and $j$-players as $k_j + 1$. Also, note that $\mathbf{k}'$ and $\mathbf{k}''$ are different variables and have no relation with $\mathbf{k}$. The primes are only to distinguish the sequence of summations: we do the computation on $\mathbf{k}''$, $\mathbf{k}'$, and finally $\mathbf{k}$.

We start from the level where $\mathbf{k}$ is given. Given neighbor configuration $\mathbf{k}$ for a focal $j$-player, its accumulated payoff can be expressed as

$$\pi_j^{\mathbf{k}} = a_{j|\mathbf{k}} + \sum_{l=1}^{n} k_l \sum_{\sum_{\ell=1}^{n} k_\ell' = k-1} \frac{(k-1)!}{\prod_{\ell=1}^{n} k_\ell'!} \left(\prod_{\ell=1}^{n} q_{\ell|l}^{k_\ell'}\right) a_{j|\mathbf{k}_{+l}'}. \tag{8}$$

The $j$-player accumulates payoff from the games organized by itself and its $\sum_{l=1}^{n} k_l = k$ neighbors. The visualization of Eq. (8) is shown in

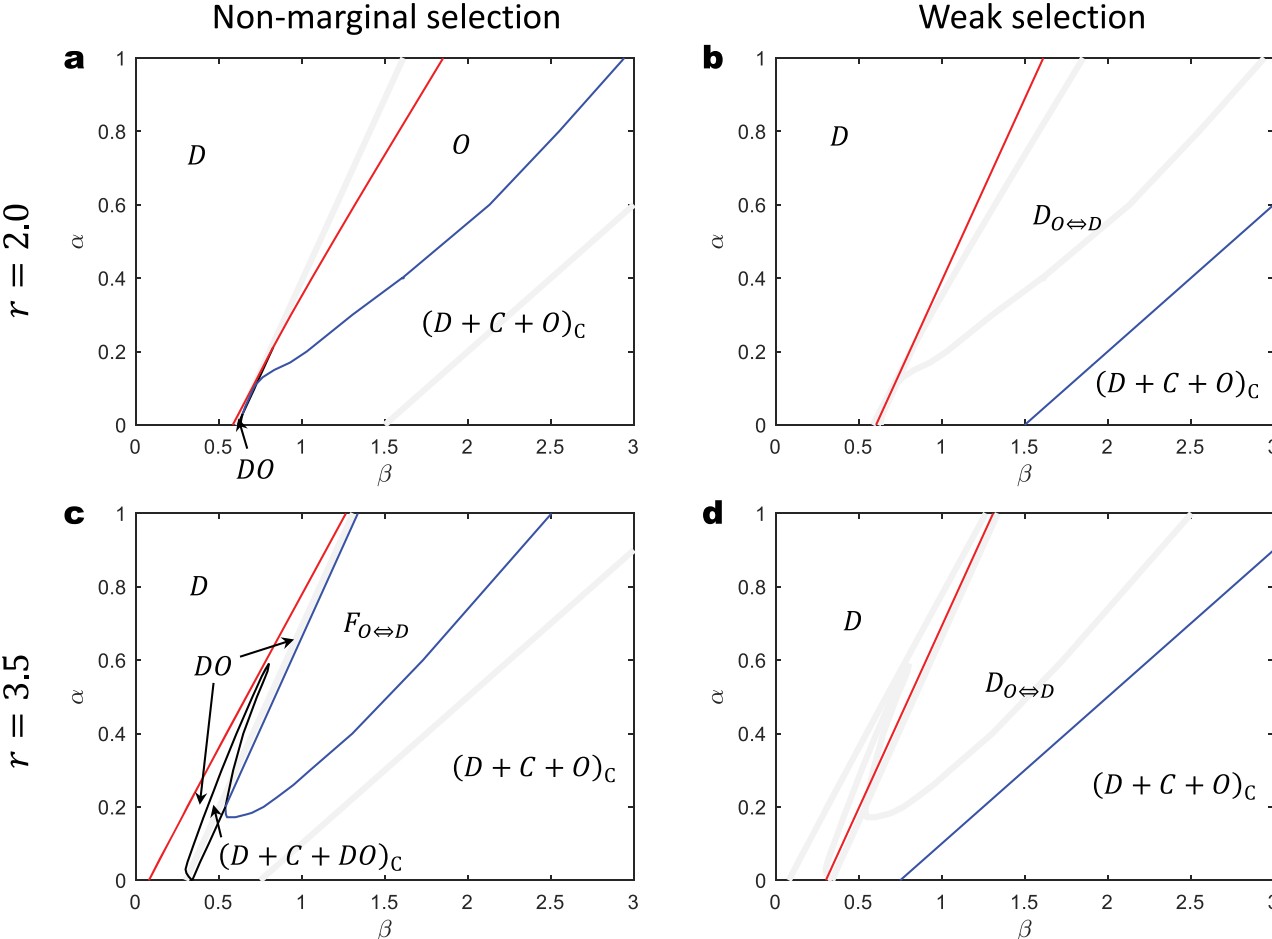

**Fig. 8 | Phase diagrams of the system behavior with pool punishment are qualitatively similar under non-marginal and weak selection strength. a** and **c** (the data are in agreement with those published in Figs. 5 and 10 from ref. 8), Numerical simulations under non-marginal selection ($\delta = 2$). The phases are defined as follows: $D$–only $D$ exists; $O$–only $O$ exists; $(D+C+O)_C$–cyclic dominance among $D$, $C$, and $O$; $F_{O\Leftrightarrow D}$–fixation of either $O$ or $D$; $DO$–$D$ and $O$ coexist; $(D+C+DO)_C$–cyclic dominance among $D$, $C$, and $DO$. **b** and **d**, The phase diagram is divided by analytical $\beta_0$ and $\beta^*$ under weak selection ($\delta \to 0^+$). Here, $\beta_0$ divides the $D$ and $D_{O\Leftrightarrow D}$ phases, while $\beta^*$ separates the $D_{O\Leftrightarrow D}$ and $(D+C+O)_C$ phases. Specifically, in **b**, $\beta_0 = 81/134 + (135/134)\alpha$ (red), $\beta^* = 3/2 + (5/2)\alpha$ (blue); in **d**, $\beta_0 = 81/268 + (135/134)\alpha$, $\beta^* = 3/4 + (5/2)\alpha$. The definition of the $D_{O\Leftrightarrow D}$ phase–the system finally evolves to full $D$ if cooperation is initially present, or to the fixation of either $O$ or $D$ in the absence of initial cooperators. **Other parameters:** $c = 1$, $k = 4$.

Fig. 9a and b. Similarly, the accumulated payoff of an $i$-player neighboring a $j$-player, given the $j$-player's neighbor configuration **k**, can be expressed and calculated by

$$\pi_{i|j}^{\mathbf{k}} = a_{i|\mathbf{k}_{-i,+j}} + \sum_{\sum_{l=1}^n k_l' = k-1} \frac{(k-1)!}{\prod_{l=1}^n k_l'!} \left( \prod_{l=1}^n q_{l|i}^{k_l'} \right)$$
$$\left( a_{i|\mathbf{k'}_{+j}} + \sum_{l=1}^n k_l' \sum_{\sum_{l'=1}^n k_{l'}'' = k-1} \frac{(k-1)!}{\prod_{l'=1}^n k_{l'}''!} \left( \prod_{\ell=1}^n q_{\ell|l}^{k_\ell''} \right) a_{i|\mathbf{k''}_l} \right). \quad (9)$$

The $i$-player accumulates payoff from the games organized by the $j$-player (Fig. 9c), itself (Fig. 9d), and its remaining $\sum_{l=1}^n k_l' = k-1$ neighbors (Fig. 9e). Further explanations of Eqs. (8) and (9) can be found in Supplementary Information.

Based on the microscopic quantities given specific **k**, we can further express expected values over all possible **k**. The expected payoff of a focal $X$-player over all possible **k** can be statistically computed as

$$\left\langle \pi_X^{\mathbf{k}} \right\rangle = \sum_{\sum_{i'=1}^n k_{i'} = k} \frac{k!}{\prod_{i'=1}^n k_{i'}!} \left( \prod_{i'=1}^n q_{i'|X}^{k_{i'}} \right) \pi_X^{\mathbf{k}}. \quad (10)$$

The possible neighbor configurations satisfying $\sum_{i'=1}^n k_{i'} = k$ are found around the $X$-player as identified by $q_{i'|X}$. Here, $i'$ is an independent count, which has no relation with $i$.

Similarly, the expected payoff of an $i$-player neighboring an $X$-player over all possible neighbor configuration $\mathbf{k}_{+i}$ of the $X$-player can be expressed as

$$\left\langle \pi_{i|X}^{\mathbf{k}_{+i}} \right\rangle = \sum_{\sum_{i'=1}^n k_{i'} = k-1} \frac{(k-1)!}{\prod_{i'=1}^n k_{i'}!} \left( \prod_{i'=1}^n q_{i'|X}^{k_{i'}} \right) \pi_{i|X}^{\mathbf{k}_{+i}}. \quad (11)$$

Here, $\mathbf{k}_{+i}$ is because we have a specific $i$-player in the neighbor configuration of the $X$-player. The remaining $k - 1$ neighbors $\sum_{i'=1}^n k_{i'} = k-1$ of the $X$-player are found around the $X$-player, as identified by $q_{i'|X}$.

Eqs. (9) and (11) may seem redundant because they do not appear directly in the final results. However, they are crucial in the process of deductions for both pairwise comparison and death-birth rules.

We also have the similar notation for expected single-game payoff. To specify, the expected payoff of an $i$-player in a single game over all co-player configuration **k**, where **k** is found neighboring

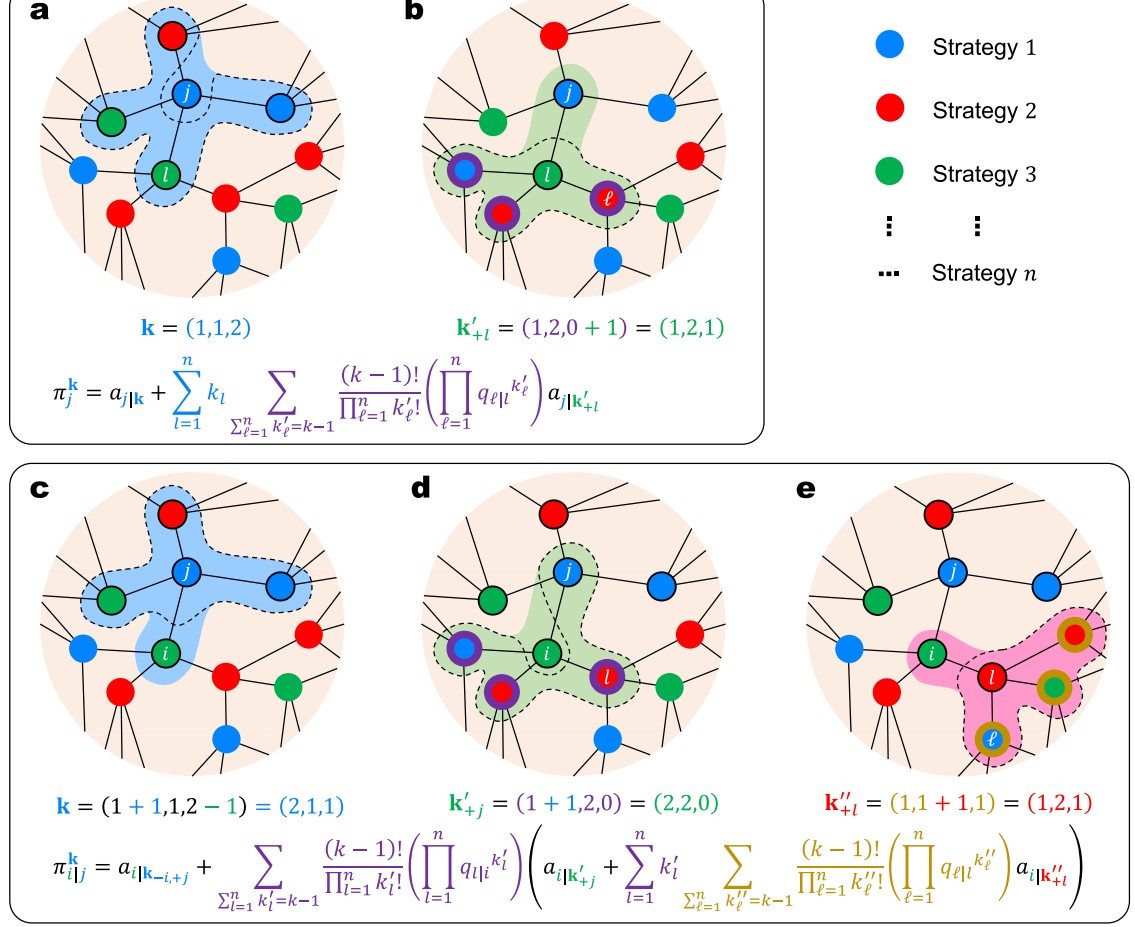

**Fig. 9 | Visualization of the bottom-up statistical payoff calculation when every individual has $k = 4$ neighbors. a** To calculate the group-based payoff of a focal $j$-player with neighbor configuration $\mathbf{k}$, we first have the game organized by the $j$-player itself, with $\mathbf{k}$ as the co-player configuration. **b** Then, we have $\sum_{l=1}^{n} k_l = k$ games organized by the $j$-player's neighbors who adopt strategy $l = 1, 2, ..., n$. For the $j$-player, the co-player configuration contains $k - 1$ undetermined co-players neighboring the $l$-player, plus the $l$-player. **c** To calculate the group-based payoff of an $i$-player neighboring a focal $j$-player, with $j$-player's neighbor configuration $\mathbf{k}$, we first have the game organized by the $j$-player. For the $i$-player, the co-player configuration is $\mathbf{k}_{-i,+j}$. **d** Then, we have the game organized by the $i$-player itself, with $k - 1$ undetermined co-players neighboring the $i$-player plus the $j$-player. **e** Finally, we have the games organized by the $i$-player's remaining $\sum_{l=1}^{n} k_l' = k - 1$ neighbors. For the $i$-player, the co-player configuration contains $k - 1$ undetermined co-players neighboring the $l$-player, plus the $l$-player.

an $X$-player, is expressed as

$$\langle a_{i|\mathbf{k}} \rangle_X = \sum_{\sum_{i'=1}^{n} k_{i'} = k} \frac{k!}{\prod_{i'=1}^{n} k_{i'}!} \left( \prod_{i'=1}^{n} q_{i'|X}^{k_{i'}} \right) a_{i|\mathbf{k}}. \tag{12}$$

The concepts in Eqs. (8)–(12) are sufficient to identify the difference between this work and the previous literature[29,33,46]. In particular, they emphasize that the minimal unit to refer is the co-player configuration $\mathbf{k}$, based on which the payoff of the multiplayer game is computed. We do not try to decompose the multi-body interaction identified by $\mathbf{k}$ into multiple pairwise interactions.

Combined with the pair approximation method[33] and detailed calculations in the strategy evolution dynamics, we can then obtain the master equation (4) in the main text (see Supplementary Note 2.4.3 for the approach of pair approximation deduction).

## The decomposition to single games

The following form of the master equation is important, which holds for any $n$-strategy multiplayer game and allows us to obtain the replicator dynamics by summing the expected payoff calculations in a

series of single games:

$$\dot{x}_i = \frac{\delta(k-2)}{2(k-1)} x_i \sum_{j=1}^{n} x_j \left( \langle a_{i|\mathbf{k}_{+j}} \rangle_i + (k-1)\langle a_{i|\mathbf{k}_{+j}} \rangle_j + \langle a_{i|\mathbf{k}_{+i}} \rangle_i - \langle a_{j|\mathbf{k}_{+i}} \rangle_j \right.$$
$$\left. - \langle a_{j|\mathbf{k}_{+i}} \rangle_i - (k-2) \sum_{j=1}^{n} x_l \langle a_{j|\mathbf{k}_{+l}} \rangle_l - \langle a_{j|\mathbf{k}_{+j}} \rangle_j \right). \tag{13}$$

In application, given the payoff functions $a_{i|\mathbf{k}}$ where $i = 1, 2, ..., n$, we can compute all $\langle \cdot \rangle$ terms and then ensemble them to obtain the replicator equations. The result of each $\langle \cdot \rangle$ should be a function of $x_1, x_2, ..., x_n$ (transformed from $q_{j|i}$ manually), degree $k$, and game parameters.

The advantage of Eq. (13) is that we have attributed everything about $\langle \cdot \rangle$ into two types, the '$\langle a_{i|\mathbf{k}_{+j}} \rangle_i$ type' and the '$\langle a_{i|\mathbf{k}_{+j}} \rangle_j$ type'. They can be expressed by matrices through $i$ and $j$:

- The $\langle a_{i|\mathbf{k}_{+j}} \rangle_i$ type:

$$
\left[ \langle a_{i|\mathbf{k}_{+j}} \rangle_i \right]_{ij} =
\begin{pmatrix}
\langle a_{1|\mathbf{k}_{+1}} \rangle_1 & \langle a_{1|\mathbf{k}_{+2}} \rangle_1 & \cdots & \langle a_{1|\mathbf{k}_{+n}} \rangle_1 \\
\langle a_{2|\mathbf{k}_{+1}} \rangle_2 & \langle a_{2|\mathbf{k}_{+2}} \rangle_2 & \cdots & \langle a_{2|\mathbf{k}_{+n}} \rangle_2 \\
\vdots & \vdots & \ddots & \vdots \\
\langle a_{n|\mathbf{k}_{+1}} \rangle_n & \langle a_{n|\mathbf{k}_{+2}} \rangle_n & \cdots & \langle a_{n|\mathbf{k}_{+n}} \rangle_n
\end{pmatrix}. \tag{14}
$$

- The $\langle a_{i|\mathbf{k}_{+j}} \rangle_j$ type:

$$
\left[ \langle a_{i|\mathbf{k}_{+j}} \rangle_j \right]_{ij} =
\begin{pmatrix}
\langle a_{1|\mathbf{k}_{+1}} \rangle_1 & \langle a_{1|\mathbf{k}_{+2}} \rangle_2 & \cdots & \langle a_{1|\mathbf{k}_{+n}} \rangle_n \\
\langle a_{2|\mathbf{k}_{+1}} \rangle_1 & \langle a_{2|\mathbf{k}_{+2}} \rangle_2 & \cdots & \langle a_{2|\mathbf{k}_{+n}} \rangle_n \\
\vdots & \vdots & \ddots & \vdots \\
\langle a_{n|\mathbf{k}_{+1}} \rangle_1 & \langle a_{n|\mathbf{k}_{+2}} \rangle_2 & \cdots & \langle a_{n|\mathbf{k}_{+n}} \rangle_n
\end{pmatrix}. \tag{15}
$$

There are $n^2$ elements in each matrix. Their diagonals are equal, meaning we can compute $n$ fewer elements. Therefore, the total amount of computation is $n^2 + n^2 - n = (2n-1)n$ elements. The computational complexity is $O(n^2)$, which can be accomplished in polynomial time (we also see that the death-birth rule's computational complexity is $O(n^3)$, as specified in Supplementary Information).

### Special linear system

Although Eq. (13) allows general calculations of any multiplayer game, we do not have to employ it directly every time. For some special payoff structures, we can deduce simplified general forms in advance. Here, we present the general results of a commonly studied subclass, the linear multiplayer games.

The linear multiplayer games in this work are defined as those whose payoff structure can be expressed as linear functions of co-player $\mathbf{k} = (k_1, k_2, \ldots, k_n)$. That is, $a_{i|\mathbf{k}} = \sum_{j=1}^{n} b_{ij} k_j + c_i$, where

$$
\mathbf{b} =
\begin{pmatrix}
b_{11} & b_{12} & \cdots & b_{1n} \\
b_{21} & b_{22} & \cdots & b_{2n} \\
\vdots & \vdots & \ddots & \vdots \\
b_{n1} & b_{n2} & \cdots & b_{nn}
\end{pmatrix},
\mathbf{c} =
\begin{pmatrix}
c_1 \\
c_2 \\
\vdots \\
c_n
\end{pmatrix}. \tag{16}
$$

For linear multiplayer games, the payoff structure is completely determined by the matrix $\mathbf{b}$ and $\mathbf{c}$.

Once we apply $a_{i|\mathbf{k}} = \sum_{j=1}^{n} b_{ij} k_j + c_i$ to compute the '$\langle a_{i|\mathbf{k}_{+j}} \rangle_i$ type' and the '$\langle a_{i|\mathbf{k}_{+j}} \rangle_j$ type' as shown in Eqs. (14) and (15) and transform all $q_{j|i}$ quantities to $x_j$ quantities, we can obtain $\langle a_{i|\mathbf{k}_{+j}} \rangle_i = (k-2) \sum_{l=1}^{n} b_{il} x_l + b_{ii} + b_{ij} + c_i$, $\langle a_{i|\mathbf{k}_{+j}} \rangle_j = (k-2) \sum_{l=1}^{n} b_{il} x_l + 2b_{ij} + c_i$. Then, substituting the results into Eq. (13) leads to Eq. (5) in the main text.

As we mentioned in the main text, in Eq. (5), $\bar{\pi}_i$ and $\bar{\pi}$ are mean payoffs of $i$-players and all players in a well-mixed population. They can be calculated by the traditional replicator dynamics, but to specify, using the matrix $\mathbf{b}$ and $\mathbf{c}$, they can also be written as $\bar{\pi}_i = k \sum_{l=1}^{n} x_l b_{il} + c_i$, $\bar{\pi} = \sum_{i=1}^{n} x_i \bar{\pi}_i = k \sum_{i=1}^{n} \sum_{l=1}^{n} x_i x_l b_{il} + \sum_{i=1}^{n} x_i c_i$. Logically, this is how we replace the corresponding terms in Eq. (5) by $\bar{\pi}_i$ and $\bar{\pi}$.

## Data availability

All data generated or analysed during this study are included within the paper and its supplementary information files.

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

## Acknowledgements

M.P. was supported by the Slovenian Research and Innovation Agency (Javna agencija za znanstvenoraziskovalno in inovacijsko dejavnost Republike Slovenije) (Grant Nos. P1-0403 and N1-0232). A.S. was supported by the National Research, Development and Innovation Office (NKFIH) under Grant No. K142948.

## Author contributions

C.W. conceived and designed the research with contributions from M.P. and A.S.; C.W. performed the calculations; C.W. and A.S. analyzed the results; C.W., M.P., and A.S. wrote the paper and approved the submission.

## Competing interests

The authors declare no competing interests.
