## [Peer Review File · Nature Communications]

Evolutionary dynamics of any multiplayer game on regular graphsREVIEWER COMMENTS

Reviewer #1 (Remarks to the Author):

In this paper, authors study evolutionary dynamics of the multi-player multi-strategy games on regular graphs, under the limit of weak selection. They derive an analytical solution of a replicator equation for this general game, taking inspiration in an analogy with the so called Ball-and-Box problem. That allows a calculation of the statistical mean payoffs of players in regular graphs. The derived analytical solution is then applied to two important models of prosocial punishments in the context of the public goods games: peer punishment and pool punishment.

The general problem studied in the paper is very complex and challenging. There have been active research and efforts in the area, but only simpler cases could be addressed so far, including the pairwise games and the multi-player games but with only two strategies. There is a crucial gap in understanding the general multi-player games with an arbitrary number of strategies, and the current paper has solved this most general case.

I find the findings from the paper outstanding and groundbreaking. The derived analytical solution would be extremely useful for researchers in the field of dynamics of human collective behaviours and complex networks.

The analyses in the paper are performed in a highly competent manner. There are extensive analytical results, all supported and validated by numerical and simulation results (including those in the Supporting Information).

Therefore, I would be happy to recommend publication of the paper in the present form. If there is a chance to revise the paper, the authors might (optionally) consider discussing existing findings from evolutionary analyses of general multi-player games with arbitrary number of strategies (similar to the goal in the paper, but for well mixed populations); see for example, i) "Analysis of the expected density of internal equilibria in random evolutionary multi-player multi-strategy games." *Journal of Mathematical Biology* 73 (2016): 1727-1760. ii) "On the expected number of equilibria in a multi-player multi-strategy evolutionary game." *Dynamic Games and Applications* 6.3 (2016): 324-346. These works analytically study the statistics of (stable) equilibrium points in multi-player games with an arbitrary number of strategies where the payoff matrix entries are random variables, showing the complexity of population dynamics increases significantly with the number of players in the game as well as that of the strategies in the population. This is highly relevant to the motivation set out in this work.

Reviewer #2 (Remarks to the Author):

In this work, the authors made a remarkable analytical achievement for understanding a class of evolutionary game dynamics on networks, i.e., multistrategy multiplayer games on networks. This class of games is important because some known major evolutionary game models, which are also socioeconomically important, belong to this class, as the authors explain in the introduction section. They developed a complete analytical framework to analyze such a model (e.g., stability and evolution of cooperation) for regular graphs (i.e. networks in which all nodes have the same degree, that is, the same number of neighbors), as Ohtsuki et al. *Nature* (2006) initially did for two-strategy two-player games on networks. Then, the present authors showcased the methods with two types of games on networks with punishment options. These two case studies (seem to) have identified new phenomena that have not been found for the same game model played in well-mixed populations. Owing to these key achievements, I think this paper can potentially reach the bar of *Nat Commun*. However, the paper is not sufficiently accessible to a large mass of audience. Furthermore, it does not successfully

convey the key points of their analytical framework, including their significance. The paper could be published in Nat Commun only when the authors could develop strong arguments to address these and other points which I highlight below. My specific feedback is as follows.

[major comments]

(1) I understand most of their analytical developments, but I do not understand how innovative the authors' present method is relative to the pair approximation by Ohtsuki et al. Nature 2006, or its direct extension to the case of multiplayer and multistage games. The authors introduce the notation of "statistical mean payoff" (e.g. line 107), which confuses me. Is this concept new? Or is it also used in Ohtsuki et al. or other similar work? In a related vein, line 285 suggests that the authors' method is different from identity-by-descent (IBD) method. Is it? If so, articulate it and highlight what is really novel in the author's method (if any, I mean, beyond directly extending Ohtsuki et al 2006 to the present case) in Introduction/Discussion. If the present methods are direct extensions of Ohtsuki et al 2006, I understand that, but the authors should say that to tell readers what their methods are really about, in light of the landscape of the research in this field. In a related vein, on line 358, they say "statistical mean payoff" and "expected payoff". Are they different? (Not just here, but throughout the manuscript.) I do not think it is fair to use the term "statistical mean payoff" without qualification.

(2) Starting line 179, two demonstrations of the proposed analytical framework were done: one with peer punishment in public goods games (which has linear payoff structure) and pool punishment in public goods games (which has nonlinear payoff structure). While I am aware that peer and pool punishment very often yield different results, which elicited various publications in the field, I need to say these two demonstrations have the same root (i.e., punishment in public goods games). For Nat Commun, I request a substantially different demonstration (i.e., not about punishment problems). If such a demonstration is successful, the paper will be more convincing in terms of the applicability of the developed methods.

(3) Fig. 2: My understanding is that panels (a) and (b) on well-mixed populations are not new, and the presently developed method, not other previous methods, enables producing (c) and (d) for the first time (I mean, analytically). If it is so, please articulate this somewhere in the main text because it is good to dissociate between what is known and what is new.

[minor comments]

(4) line 80: "On a regular graph,". The statement of this sentence also holds true for non-regular graphs.

(5) line 84 and elsewhere: The authors emphasize the connection of their analytical approach to "Ball-and-Box problem", but it is trivial and this association is not really a large contributor to their analytical development. I encourage the authors to de-emphasize it.

(6) line 96: "strength" -> "limit"

(7) line 120: "payoff level". Here and there, the term "level" is vaguely used. At least I do not understand what this means.

(8) line 137: "infinitely population" -> "infinite population, i.e.," for example.

(9) line 210: "The analytical results align qualitatively with ... previous research [49], as shown in Fig. 3." If the previous research shows (qualitatively) the same phase diagram, what is the point of showing Fig. 3? Is it just a confirmation? Or are the authors claiming that their approach is analytical whereas the previous ones were only numerical? Please clarify. Same for lines 254-255 on Fig. 5.

(10) lines 265-266: This opening sentence of the Discussion section is too hard to follow and not accurate either. "The essence of ... is" is an overstatement. This is not the only essence. Does "spatial" include the case of general networks? "marginal game effect" is vague throughout the paper.

(11) line 274: "fills" -> "fill"

(12) line 295: "Nonetheless, ..." With this sentence, the authors are repeating the sentence just before this sentence. Remove?

(13) line 339: "k' and k" are independent of k". I am confused. There is no notion of statistical independence here? What does "independence" mean?

(14) lines 395, 402, and 410: I suggest moving these three Methods subsections to the SI.

(15) line 418: What does \sim mean?

Response to Reviewer #1

In this paper, authors study evolutionary dynamics of the multi-player multi-strategy games on regular graphs, under the limit of weak selection. They derive an analytical solution of a replicator equation for this general game, taking inspiration in an analogy with the so called Ball-and-Box problem. That allows a calculation of the statistical mean payoffs of players in regular graphs. The derived analytical solution is then applied to two important models of prosocial punishments in the context of the public goods games: peer punishment and pool punishment.

The general problem studied in the paper is very complex and challenging. There have been active research and efforts in the area, but only simpler cases could be addressed so far, including the pairwise games and the multi-player games but with only two strategies. There is a crucial gap in understanding the general multi-player games with an arbitrary number of strategies, and the current paper has solved this most general case.

I find the findings from the paper outstanding and groundbreaking. The derived analytical solution would be extremely useful for researchers in the field of dynamics of human collective behaviours and complex networks. The analyses in the paper are performed in a highly competent manner. There are extensive analytical results, all supported and validated by numerical and simulation results (including those in the Supporting Information).

We thank our Reviewer for this supporting report and the constructive suggestions. Our replies to the raised points are typed in blue.

Therefore, I would be happy to recommend publication of the paper in the present form. If there is a chance to revise the paper, the authors might (optionally) consider discussing existing findings from evolutionary analyses of general multi-player games with arbitrary number of strategies (similar to the goal in the paper, but for well mixed populations); see for example, i) “Analysis of the expected density of internal equilibria in random evolutionary multi-player multi-strategy games.” *Journal of Mathematical Biology* 73 (2016): 1727-1760. ii) “On the expected number of equilibria in a multi-player multi-strategy evolutionary game.” *Dynamic Games and Applications* 6.3 (2016): 324-346. These works analytically study the statistics of (stable) equilibrium points in multi-player games with an arbitrary number of strategies where the payoff matrix entries are random variables,

showing the complexity of population dynamics increases significantly with the number of players in the game as well as that of the strategies in the population. This is highly relevant to the motivation set out in this work.

Indeed, these are relevant works to the concept we studied here. Therefore, we are happy to integrate them into the motivation section. Thank you for your recommendation!

Response to Reviewer #2

In this work, the authors made a remarkable analytical achievement for understanding a class of evolutionary game dynamics on networks, i.e., multi-strategy multiplayer games on networks. This class of games is important because some known major evolutionary game models, which are also socioeconomically important, belong to this class, as the authors explain in the introduction section. They developed a complete analytical framework to analyze such a model (e.g., stability and evolution of cooperation) for regular graphs (i.e. networks in which all nodes have the same degree, that is, the same number of neighbors), as Ohtsuki et al. Nature (2006) initially did for two-strategy two-player games on networks. Then, the present authors showcased the methods with two types of games on networks with punishment options. These two case studies (seem to) have identified new phenomena that have not been found for the same game model played in well-mixed populations. Owing to these key achievements, I think this paper can potentially reach the bar of Nat Commun. However, the paper is not sufficiently accessible to a large mass of audience. Furthermore, it does not successfully convey the key points of their analytical framework, including their significance. The paper could be published in Nat Commun only when the authors could develop strong arguments to address these and other points which I highlight below. My specific feedback is as follows.

We thank our Reviewer for detailed comments on our work. Our replies to the raised points are typed in blue.

[major comments]

(1) I understand most of their analytical developments, but I do not understand how innovative the authors' present method is relative to the pair approximation by Ohtsuki et al. Nature 2006, or its direct extension to the case of multiplayer and multistage games. The authors introduce the notation of "statistical mean payoff" (e.g. line 107), which confuses me. Is this concept new? Or is it also used in Ohtsuki et al. or other similar work? In a related vein, line 285 suggests that the authors' method is different from identity-by-descent (IBD) method. Is it? If so, articulate it and highlight what is really novel in the author's method (if any, I mean, beyond directly extending Ohtsuki et al 2006 to the present case) in Intro-

duction/Discussion. If the present methods are direct extensions of Ohtsuki et al 2006, I understand that, but the authors should say that to tell readers what their methods are really about, in light of the landscape of the research in this field. In a related vein, on line 358, they say “statistical mean payoff” and “expected payoff”. Are they different? (Not just here, but throughout the manuscript.) I do not think it is fair to use the term “statistical mean payoff” without qualification.

Ohtsuki et al. (Nature 2006) used pair approximation to study a two-strategy two-player game and found the famous $b/c > k$ law. We did not expand “directly” on this very first work because, as we stated in the Introduction and Discussion, this branch developed gradually in the past decades, for examples to multi-strategy two-player games and two-strategy multiplayer games. In our case, however, we complete the last piece of the puzzle by solving multi-strategy multiplayer games.

We stress that the algorithms we have developed on “statistical mean payoff” are new, due to group-based interactions in multiplayer games. As we pointed out in the Discussion and Methods sections, we do not try to further decompose the payoff calculation into the outcome of pairwise interactions, as Ohtsuki et al. did in two-player games. Instead, the smallest irreducible unit in our payoff calculation is the group consisting of the focal player and its k co-players. Our group-based payoff calculation for an arbitrary number of strategies and the discussion on their computational complexity have not been studied by previous research. We try to highlight this further in the revised manuscript.

To avoid any confusion caused by the term “statistical mean payoff”, we have modified the title of this subsection to “Group-based payoff with any number of strategies”. This intuitively highlights the groundbreaking point of our work to previous research: the theoretical expression of group-based payoff with any number of strategies.

According to our intention, both “statistical mean payoff” and “expected payoff” are some kind of expected payoff, but they refer to different stages in computation. In the original manuscript, we used the term “statistical mean payoff” to represent the expected payoff from a single game, and used the term “expected payoff” to represent the expected value of accumulated payoff. The relationship between them is that the accumulated payoff is the sum of $k + 1$ single-game payoffs obtained in groups centered on oneself and one’s neighbors (but the actual calculation of them is affected by condi-

tional probabilities based on known and unknown strategies of co-players). To distinguish these two concepts more clearly, we now cancel the original terms and directly write them as “expected accumulated payoff” and “expected single-game payoff”. Meanwhile, we add more explanations to further emphasize the relationship between them.

The pair approximation method we use is different from the IBD method. The latter belongs to a different methodological family on the same models. The conclusions we obtained happen to be similar to the ones previously obtained by the IBD method, thus we mention and discuss it. However, we realize that a technical term like “IBD” may be cryptic to the general reader. In the revised manuscript, we choose not to mention this terminology directly, but keep the core message we wanted to convey: “Our findings are consistent with those obtained by a different approach” rather than “Our findings are consistent with those obtained by the IBD method”.

(2) Starting line 179, two demonstrations of the proposed analytical framework were done: one with peer punishment in public goods games (which has linear payoff structure) and pool punishment in public goods games (which has nonlinear payoff structure). While I am aware that peer and pool punishment very often yield different results, which elicited various publications in the field, I need to say these two demonstrations have the same root (i.e., punishment in public goods games). For *Nat Commun*, I request a substantially different demonstration (i.e., not about punishment problems). If such a demonstration is successful, the paper will be more convincing in terms of the applicability of the developed methods.

The reason we prefer to present both the peer and pool punishment in the main text is that they represent two payoff structures that are conceptually different, as our Reviewer also acknowledged. In particular, these two different payoff structures correspond to two formulas that we developed. The peer punishment with a linear payoff structure demonstrates the ease of implementation of the simplified formula we obtained, while the pool punishment with a nonlinear payoff structure shows our general solution’s feasibility for any payoff structure. Thus, we believe that showing these two demonstrations in the main text is sufficient to demonstrate the applicability in different aspects of our general framework.

Importantly, the general framework we developed can be applied to other games with different mechanisms, as we claimed in the original manuscript.

Therefore, we accept your suggestion and significantly expand the Supplementary Information, which increases the workload of the manuscript while keeping the main text concise. Accordingly, we add two new demonstrations to best improve the quality of our paper. In particular, we add (1) public goods games with the reward mechanism (the number of strategies $n = 3$) in subsection S3.4 and (2) the multi-stage public goods game (the number of strategies $n = 4$) in subsection S3.5. These models are completely different from the punishment problem and thus illustrate the broad applicability of our method. We mention them in the end of the Discussion section. The comparison of our new results with previous numerical simulations also confirm our main conclusion, namely the qualitative match between diagrams of spatial populations obtained for under different selection strengths.

(3) Fig. 2: My understanding is that panels (a) and (b) on well-mixed populations are not new, and the presently developed method, not other previous methods, enables producing (c) and (d) for the first time (I mean, analytically). If it is so, please articulate this somewhere in the main text because it is good to dissociate between what is known and what is new.

As the title of our paper highlights, the key contribution of our work is the development of a technique that can handle multi-strategy multiplayer games in structured populations. This technique, which produces panels (b) and (d), has not been reported before. Indeed, the mathematical techniques for producing (a) and (c) were previously available. However, the results in (a) and (c) were never published. Furthermore, they are crucial to illustrate the relation between the system behavior obtained in well-mixed and in structured populations. In the revised manuscript, we further clarify the difference between new and existing techniques that generate different panels in Figs. 2 and 4.

[minor comments]

(4) line 80: “On a regular graph,”. The statement of this sentence also holds true for non-regular graphs.

Our original phrase may be unclear but we also cannot fully agree with the suggested point. We wrote: “On a regular graph, the number of co-players in every multiplayer game is equivalent to the number of neighbors.”

However, the same statement cannot hold for non-regular graphs. When participating a game organized by one's neighbors, the number of co-players would be the number the organizer's neighbors, which is not necessarily equal to the number of the focal player's neighbors.

To better convey the intended information and keep the fluent logic of statement, we change this sentence to: "On a regular graph, the number of co-players in every multiplayer game is equivalent to the constant number k of neighbors."

(5) line 84 and elsewhere: The authors emphasize the connection of their analytical approach to "Ball-and-Box problem", but it is trivial and this association is not really a large contributor to their analytical development. I encourage the authors to de-emphasize it.

We are sorry if our original description was misleading to the Reviewer. Indeed, this analogy does not work directly in the analysis in evolutionary dynamics on graphs. However, it is helpful in intuitively understanding the strategy configuration for a single multi-strategy multiplayer game before we start to discuss the evolutionary dynamics. We believe this could be valuable for the general readership of Nature Communications from interdisciplinary background. In the revised manuscript, we carefully consider the role of "Balls-and-Boxes problem", which is done before we expand the main theory. We only keep it for an introduction purpose but remove it in the Discussion section.

(6) line 96: "strength" \rightarrow "limit"

Corrected. Thank you!

(7) line 120: "payoff level". Here and there, the term "level" is vaguely used. At least I do not understand what this means.

The "level" in the original manuscript refers to a stage in calculation; for example, "the accumulated payoff level" "the single game level" etc. In the revised manuscript, we remove "level" in similar places accordingly.

(8) line 137: "infinitely population" \rightarrow "infinite population, i.e.," for example.

Sorry for this flaw and we correct it in the revised manuscript. Thank you.

(9) line 210: “The analytical results align qualitatively with ... previous research [49], as shown in Fig. 3.” If the previous research shows (qualitatively) the same phase diagram, what is the point of showing Fig. 3? Is it just a confirmation? Or are the authors claiming that their approach is analytical whereas the previous ones were only numerical? Please clarify. Same for lines 254-255 on Fig. 5.

We are sorry if our original statement was unclear. The presented analytical approach was completely missing in the previous literature. Therefore, we are interested in comparing the previous numerical simulations with our analytical calculations and show their consistency. As we write, they consistently reveal unique phases that only exist in structured populations. We further clarify this in the revised manuscript, which may help to connect research activities along these independent paths: “We also compare the analytical predictions by our framework to the results from previous work, which was only at a numerical level.”

(10) lines 265-266: This opening sentence of the Discussion section is too hard to follow and not accurate either. “The essence of ... is” is an overstatement. This is not the only essence. Does “spatial” include the case of general networks? “marginal game effect” is vague throughout the paper.

We have modified this criticized part. “Spatial” includes general networks as well, but it is not important here. In this paragraph, we aim to state the reason why evolutionary dynamics under weak selection can be analytically feasible.

(11) line 274: “fills” → “fill”

Again, we are sorry for such flaws. We have corrected it in the revised version.

(12) line 295: “Nonetheless, ...” With this sentence, the authors are repeating the sentence just before this sentence. Remove?

Indeed, this was a redundant statement. We have corrected it.

(13) line 339: “ \mathbf{k}' and \mathbf{k}'' are independent of \mathbf{k} ”. I am confused. There is no notion of statistical independence here? What does “independence” mean?

We use \mathbf{k} , \mathbf{k}' , and \mathbf{k}'' to represent three independent variables, which keeps the calculation within an equation simple to follow. We could also use \mathbf{k} , \mathbf{j} , \mathbf{l} , \mathbf{m} , and so on, but which would increase the difficulty for readers to remember the sequence in calculation. However, the Reviewer’s question reminds us that the current expression could be misleading. Therefore, we now directly state that they are all independent variables.

(14) lines 395, 402, and 410: I suggest moving these three Methods subsections to the SI.

Indeed, some information in these subsections could be shifted to the SI, highlighting the key information of the main text. However, some information needs to be integrated to the main text to ensure it self-contained. We carefully reorganize the information in these Methods subsections and shift them to the best place that we believe.

In particular, we refine the definition and discussion of peer and pool punishment in the main text and SI by integrating the information in the Methods subsections. On the other hand, the Methods subsection of death-birth is completely removed, because the SI is sufficient to describe it. The corresponding places that readers should refer to are adjusted in the main text when mentioning death-birth.

(15) line 418: What does \sim mean?

It means equivalent or proportional. We attempted to avoid writing the constant coefficient $\delta(k-1)/k$ (for death-birth) in the original equation, because this coefficient does not affect the system’s property about equilibrium and stability. For simplicity, we only convey the key information (i.e., the left side is proportional to something).

The Reviewer’s question reminds us that the symbol \sim could be misleading. Therefore, we change \sim to \propto , which is the formal symbol representing “proportional”. Consequently, the original text should be changed

to $\dot{x}_i \propto x_i(\pi_i^{(0)} - \pi_i^{(2)})$. While this part no longer exists due to your point (14), similar places, such as $\dot{x}_i \propto x_i(\pi_i^{(0)} - \pi_i^{(1)})$ and $\dot{x}_i \propto x_i(\bar{\pi}_i - \bar{\pi})$, are also corrected accordingly, with short explanations of the symbol.

REVIEWERS' COMMENTS

Reviewer #2 (Remarks to the Author):

The authors did diligent work to successfully address all my concerns. Especially, the additional analysis of two example systems (i.e., item "(2)" in the response letter) is convincing and broadens the applicability and generality of the proposed analytical framework. Well done. I recommend the publication of this paper.